

# The $^{18}$O ecohydrology of a grassland ecosystem – predictions and observations

Regina T. Hirl[1,4], Hans Schnyder[1], Ulrike Ostler[1], Rudi Schäufele[1], Inga Schleip[1,2], Sylvia H. Vetter[3], Karl Auerswald[1], Juan C. Baca Cabrera[1], Lisa Wingate[4], Margaret M. Barbour[5], Jérôme Ogée[4]

[1]Lehrstuhl für Grünlandlehre, Technische Universität München, 85354 Freising, Germany
[2]Nachhaltige Grünlandnutzungssysteme und Grünlandökologie, Hochschule für nachhaltige Entwicklung Eberswalde, 16225 Eberswalde, Germany
[3]The School of Biological Sciences, University of Aberdeen, Aberdeen AB24 3UU, UK
[4]UMR ISPA, INRA, 33140 Villenave d'Ornon, France
[5]Sydney Institute of Agriculture, The University of Sydney, NSW 2570, Sydney, Australia

*Correspondence to*: Regina Hirl (regina.hirl@tum.de)

**Abstract.** The oxygen isotope composition ($\delta^{18}$O) of leaf water ($\delta^{18}$O$_{leaf}$) is an important determinant of environmental and physiological information found in biological archives, but the system-scale understanding of the propagation of the $\delta^{18}$O of rain through soil and xylem water to $\delta^{18}$O$_{leaf}$ has not been verified for grassland. Here we report a unique and comprehensive dataset of biweekly $\delta^{18}$O observations in soil, stem and leaf waters made over seven growing seasons in a temperate, drought-prone, mixed-species grassland. Using an $^{18}$O-enabled soil-plant-atmosphere transfer model, we evaluated our ability to predict the dynamics of $\delta^{18}$O in soil water, the depth of water uptake, and the effects of soil and atmospheric moisture on $^{18}$O-enrichment of leaf water ($\Delta^{18}$O$_{leaf}$) in this ecosystem. The model accurately predicted the $\delta^{18}$O dynamics of the different ecosystem water pools. Water uptake occurred from shallow soil depths throughout dry and wet periods in all years, presumably because of the high grazing pressure. $\Delta^{18}$O$_{leaf}$ responded to both soil and atmospheric moisture and was best described when leaf water was separated into two non-mixing water pools. The close agreement between model predictions and observations is remarkable (and promising) as model parameters describing the relevant physical features or functional relationships of soil and vegetation were held constant with one single value for the entire mixed-species ecosystem.

## 1 Introduction

The stable oxygen isotope composition ($\delta^{18}$O) of meteoric water varies greatly in space and time. During rainfall, meteoric water imparts its isotopic signal to soil water, but this signal ($\delta^{18}$O$_{rain}$) is subsequently altered during mixing and other fractionating processes occurring within the soil profile. The oxygen isotope composition of leaf water ($\delta^{18}$O$_{leaf}$) differs strongly from $\delta^{18}$O$_{rain}$ as leaves undergo an isotopic enrichment during transpiration. As a consequence, $\delta^{18}$O$_{leaf}$ carries important environmental and physiological information that is imprinted on photosynthetic products and archived in long-lived cellular compounds such as cellulose in tree rings (Farquhar et al., 2007; Barbour, 2007; Treydte et al., 2014; Lehmann



et al., 2018). The $\delta^{18}O$ of leaf water also imprints the oxygen isotope compositions of atmospheric $CO_2$ and molecular oxygen, a property that can be used to estimate regional and global scale land primary productivity from seasonal to millennium time scales (Dole et al., 1954; Farquhar et al., 1993; Bender et al., 1994; Luz and Barkan, 2011; Wingate et al., 2009; Welp et al., 2011). A quantitative understanding of the hydrological and plant morpho-physiological mechanisms

controlling $\delta^{18}O_{leaf}$ is therefore fundamental to biological, Earth and environmental science disciplines (Barbour, 2007).

Studies that deal with the $\delta^{18}O$ in water and biomass compartments of grassland, the largest terrestrial biome after forest, are sparse (e.g. Flanagan and Farquhar, 2014; Webb and Longstaffe, 2003, 2006; Ramirez et al., 2009; Riley et al., 2002, 2003). To our knowledge, simultaneous observations of seasonal variations of the isotopic composition of the different water pools in a temperate grassland ecosystem over multiple years have not been reported so far. Only datasets covering short periods

(e.g. Lai et al., 2008; Leng et al., 2013) or one single vegetation period (e.g. Wen et al., 2012) have been reported. In addition, our system-scale understanding of the propagation of the rainwater $\delta^{18}O$ signal through soil water and plant xylem water to the leaf water has as yet not been verified for grassland. As a consequence, our quantitative knowledge of the drivers of $\delta^{18}O_{leaf}$ in grassland ecosystems is limited.

The isotopic composition of the plant source water ($\delta^{18}O_{xylem}$, henceforth equated with $\delta^{18}O_{stem}$) can vary over time through

changes in the depth of soil water uptake by roots or direct changes in soil water isotopic composition. For example, summer rains are often isotopically distinct ($^{18}O$-enriched) relative to winter precipitation, generating variations of $\delta^{18}O_{soil}$ ($\delta^{18}O$ of soil water) with soil depth. Apart from the temporal distribution of rainfall amounts and associated $\delta^{18}O_{rain}$, the relationship between $\delta^{18}O_{rain}$ and $\delta^{18}O_{stem}$ is affected by soil properties (that determine water storage, transport and mixing of rainwater with water stored at depth in the soil profile), the depth distribution of roots and their specific activities and atmospheric

conditions and vegetation properties (that determine transpiration, and soil evaporation and associated enrichment of $\delta^{18}O_{soil}$ near the soil surface). Assuming that root water uptake proceeds without $^{18}O$ discrimination (Dawson et al., 2002), the comparison of $\delta^{18}O_{soil}$ and $\delta^{18}O_{stem}$ can help identify the depth of root water uptake (e.g. Durand et al., 2007) and how it changes during drought (e.g. Hoekstra et al., 2014; Nippert and Knapp, 2007a). So far, studies on potential shifts of root water uptake depth in $C_3$ grassland communities during drought were mainly conducted using rainout shelters and comparing

the water uptake depth in droughted and control plots (Hoekstra et al., 2014; Prechsl et al., 2015). Thus it is still unclear how edaphic drought arising under natural conditions modifies the root water uptake depth in $C_3$ grassland communities over time, especially at a multi-seasonal timescale.

The mechanisms driving the enrichment of leaf water can be studied separately from those driving changes in $\delta^{18}O_{stem}$ by expressing all isotopic compositions as enrichments above $\delta^{18}O_{stem}$, i.e., $\Delta^{18}O_{leaf} = \delta^{18}O_{leaf} - \delta^{18}O_{stem}$. The process of

evaporative enrichment was first modelled by Craig and Gordon (1965) for open water bodies and adapted to leaves by Dongmann et al. (1974). Many authors have since noted a discrepancy between the $^{18}O$ enrichment at the evaporative sites predicted by the Craig-Gordon model ($\Delta^{18}O_e$) and leaf water enrichment ($\Delta^{18}O_{leaf}$). This discrepancy has been interpreted conceptually with two different models called "two-pool" model (Leaney et al., 1985; Yakir et al., 1994) and "Péclet" model (Farquhar and Lloyd, 1993; Farquhar et al., 2007). In the two-pool model, leaf water is assumed compartmentalised between



evaporatively $^{18}$O-enriched water (supposed to represent mainly mesophyll cells) and un-enriched water (supposed to represent veins and associated ground tissues). In the so-called Péclet model, the mixing of water isotopes within the leaf lamina is assumed incomplete because of a limited back diffusion of heavy water from the evaporative sites to the remaining leaf lamina as a result of the high tortuosity of the path of water within the mesophyll. This incomplete mixing is characterised by a Péclet number $p$, defined as the ratio of advection to back-diffusion (Farquhar and Lloyd 1993; Cuntz et al., 2007). The two models predict a different effect of transpiration rate on the proportional difference ($\varphi$) between the $^{18}$O enrichment predicted by the Craig-Gordon model and the observed $^{18}$O enrichment of leaf water: $\varphi = 1 - \Delta^{18}O_{leaf}/\Delta^{18}O_e$ (Song et al., 2013; Cernusak et al., 2016). Because $\Delta^{18}O_{leaf}$, rather than $\Delta^{18}O_e$, imprints sugars (Barbour et al. 2000; Cernusak et al. 2003) and ultimately organic matter (Barbour and Farquar 2000; Helliker and Ehleringer 2002; Barbour 2007), the choice of the model relating $\Delta^{18}O_{leaf}$ and $\Delta^{18}O_e$ has important implications. The Péclet model predicts an increase of $\varphi$ with leaf transpiration while in the two-pool model $\varphi$ does not respond to transpiration and is expected to be constant, at least on short (hourly to daily) timescales. Thus far, experimental and empirical studies on a large range of plant species have provided mixed results on these two alternative models of $\Delta^{18}O_{leaf}$, with some studies supporting the two-pool model and others the Péclet model (e.g. Barbour et al., 2000, 2004; Loucos et al., 2015; Song et al., 2015; Cernusak et al., 2016). The question as to which model is more appropriate for predicting the $\Delta^{18}O$ of grassland canopy-scale leaf water is particularly relevant for the modelling of $\Delta^{18}O_{leaf}$, and ultimately $\delta^{18}O_{leaf}$, at larger temporal and spatial scales.

In general, $\Delta^{18}O_{leaf}$ responds strongly to changes in atmospheric humidity or the isotope composition of water vapour (e.g. Farquhar et al., 2007) and to changes in stomatal conductance (Wang and Yakir, 1995; Barbour and Farquhar, 2000; Helliker and Ehleringer, 2000; Xiao et al., 2012). However, it is generally not known whether edaphic drought, *via* its effect on stomatal conductance, indirectly affects the relative humidity response of leaf water enrichment. To our knowledge, the only study that reports a distinct effect of edaphic drought on $\Delta^{18}O_{leaf}$ is that of Ferrio et al. (2012) on *Vitis vinifera*. Based on their results, and theoretical considerations regarding the effect of soil water availability on leaf stomatal closure and energy budget and associated $^{18}$O fractionation, one would expect a positive effect of edaphic drought on leaf water enrichment. Yet, whether or not drought exerts a measurable effect on $\Delta^{18}O_{leaf}$ of grasslands, often found in climates with sporadic or prolonged drought periods, is not known.

The interpretation of the isotopic composition of water from samples collected in natural ecosystems is complicated by the fact that multiple environmental, as well as site or plant morpho-physiological factors vary simultaneously, causing difficulties in disentangling the effect of different parameters on the water isotope composition. Hence, process-based ecosystem-scale models are key to aid the interpretation of the water isotope signals in response to environmental and morphological parameters (e.g. Riley et al., 2003). Here we evaluate our system-scale eco-hydrological understanding of the propagation of the $\delta^{18}O$ signal of rainwater through soil water pools, root water uptake and $^{18}$O enrichment of leaf water in a drought-prone grassland ecosystem. For this, we systematically trace, predict and validate $\delta^{18}O_{soil}$, $\delta^{18}O_{stem}$ and $\Delta^{18}O_{leaf}$ and evaluate their sensitivity to input parameters. Specifically, we ask: what is the plant community's depth of root water uptake and does it shift in response to soil water scarcity? Is the two-pool model or the Péclet model more appropriate for describing





$\Delta^{18}O_{leaf}$ at the canopy scale? Does $\Delta^{18}O_{leaf}$ respond to edaphic drought in grasslands? And more generally: what is the sensitivity of soil, stem and leaf water $\delta^{18}O$ to changes in soil and vegetation parameters that are suspected to alter ecosystem water dynamics? To explore these questions we compared predictions from the $^{18}O$-enabled soil-plant-atmosphere model MuSICA (Ogée et al., 2003; Wingate et al., 2010; Gangi et al., 2015) with those observed in a unique, multi-annual data set

(7 years) of growing season, biweekly samplings and $\delta^{18}O$ analysis of soil water (at two depths), mixed-species stem and midday leaf water, atmospheric water vapour, along with rainfall amount and $\delta^{18}O_{rain}$ data.

## 2 Materials and Methods

### 2.1 Study site

The study was performed inside pasture paddock no. 8 of Grünschwaige Grassland Research Station near Freising, Germany

(Schnyder et al., 2006). Mean annual air temperature from 2006 to 2012 was 9.3°C, and mean annual precipitation was 743 mm, as measured at Munich airport meteorological station 3 km from the field site. The soil is a Mollic Fluvisol, with a shallow topsoil of low water holding capacity (66 mm plant available field capacity) overlying coarse calcareous gravel. The depth to the groundwater table is around 1.5 m.

During the main vegetation period (mid-April to beginning of November) the paddock was grazed continuously by Limousin

suckler cows (Schnyder et al., 2006). Animal stocking density was adjusted periodically to maintain a constant sward height of about 7 cm. This management system aimed at maintaining a constant sward state by continuously balancing pasture grass production and consumption by the grazing cattle.

### 2.2 Sampling

Precipitation water was collected following events during the vegetation periods of 2007 to 2012, and during winter

2007/2008 (see Supplement, Methods S1). Leaf, stem, soil, groundwater and atmospheric moisture samples were collected on non-rainy days, between 11 am and 4 pm CEST (Central European Summer Time). Sampling occurred at approximately biweekly intervals during the vegetation periods from April 2006 to September 2012. Samples were collected at random locations in an area of about 1 ha in the vicinity of an eddy flux tower installed near the centre of the paddock. On each date, two replicate samples of leaf, (pseudo-)stem and soil were collected. Soil samples were taken at two depths (7 and 20 cm)

using an auger. Leaf and stem samples were obtained as mixed-species collections of the co-dominant species: four $C_3$ grasses (*Lolium perenne*, *Poa pratensis*, *Phleum pratense*, *Dactylis glomerata*), one rosette dicot (*Taraxacum officinale*) and one legume (*Trifolium repens*). Each leaf sample was comprised of the integral youngest fully-expanded and mature (but not senescing) leaf blades of two vegetative tillers of *D. glomerata* and 16 vegetative tillers of *L. perenne*, *P. pratensis* and *P. pratense*, one half of a leaf blade of *T. officinale* (with the latter severed along, but not including, the mid-vein) and two

trifoliate leaves of *T. repens*. Stem (xylem) samples comprised the mid-vein of *T. officinale*, the petioles of the two *T. repens*



leaves and the basal part of the vegetative grass tillers, except for the outer-most part that was removed as it could have been subject to evaporative enrichment [cf. pseudo-stem in Fig. 1 of Liu et al. (2017)].

Atmospheric moisture was collected by pumping ambient air through a glass coil immersed in a dry ice-ethanol mixture at a flow rate of 1 L min$^{-1}$ over periods of 2-6 h around noon. Groundwater was sampled from a well located at about 100 m
upstream of the ground water flow beneath paddock no. 8.

All plant and soil samples were immediately transferred to 12 mL Exetainer vials (Labco, High Wycombe, UK), sealed and covered with Parafilm. All samples were stored in a freezer at approx. -18°C until water extraction using a cryogenic vacuum distillation apparatus (Liu et al., 2016).

## 2.3 Isotope analysis

Oxygen isotope composition was expressed in per mil (‰) deviation relative to a standard:

$$\delta^{18}O = (R_{sample}/R_{standard} - 1), \tag{1}$$

where $R_{sample}$ and $R_{standard}$ are the $^{18}O/^{16}O$ ratios of the sample and the V-SMOW standard (Vienna Standard Mean Ocean Water). Samples collected between 2007 and 2012 were analysed by Cavity Ring-Down Spectroscopy using previously described procedures (Liu et al., 2016). Water samples collected in 2006 were analysed with an IsoPrime isotope ratio mass
spectrometer interfaced with a multi-flow equilibration unit (both GVI, Manchester, UK). Each sample was measured against a laboratory standard gas, which was previously calibrated against secondary isotope standards (V-SMOW, V-SLAP and V-GISP). Heavy and light laboratory water standards, that spanned the range of $\delta^{18}O$ values in the dataset, were analysed every five samples. Analytical uncertainty was 0.2‰. $\delta^{18}O$ measurements obtained by Cavity Ring-Down Spectroscopy were linearly related with those obtained by isotope ratio mass spectrometry ($n = 176$; $R^2 = 0.99$). In a previous study, we
found no difference between the results from spectroscopy-based and pyrolysis-based measurements performed using a TC/EA HTC coupled to an isotope ratio mass spectrometer (see Liu et al., 2017).

## 2.4 MuSICA modelling

The isotope-enabled soil-plant-atmosphere model MuSICA (Ogée et al., 2003; Wingate et al., 2010; Gangi et al., 2015) was parameterised for the studied grassland based on data collected at the site or taken from the literature (for details and
parameter values, see below and Supplement, Methods S2 and Table S1) and validated using eddy flux data from the same site (Fig. S1).

### 2.4.1 Meteorological forcing and iso-forcing

MuSICA was forced by half-hourly values of meteorological data and $\delta^{18}O$ of water vapour ($\delta^{18}O_{vapour}$) and rainwater ($\delta^{18}O_{rain}$). Wind speed, precipitation, air temperature, relative humidity and air pressure data were obtained from the Munich
airport meteorological station, located at about 3 km south of the experimental site. Radiation was calculated as the mean of





two weather stations located 10 km west and 12 km east of the experimental site. $CO_2$ concentration was measured at the site using an open-path infrared $CO_2/H_2O$ gas analyser (LI-7500, LI-Cor, Lincoln, USA). Observations of $\delta^{18}O_{vapour}$ and $\delta^{18}O_{rain}$ at the experimental site were used as forcing variables in MuSICA. If unavailable, $\delta^{18}O_{vapour}$ and $\delta^{18}O_{rain}$ estimates were obtained from globally-gridded reconstructions derived from the isotope-enabled, nudged atmospheric general circulation

model IsoGSM (Yoshimura et al., 2011). The predicted $\delta^{18}O_{vapour}$ and $\delta^{18}O_{rain}$ at the grid point relevant to our site were first corrected for their offset with observed data, as predictions were found to be more enriched by 2‰ and 1.3‰ on average compared to the $\delta^{18}O_{vapour}$ and $\delta^{18}O_{rain}$ measured at the site (Figs. S2–S4).

### 2.4.2 Soil parameters

Soil structural properties (proportion of quartz and organic matter) as well as hydraulic characteristics (water retention and

hydraulic conductivity) were determined on soil core samples taken at the site (Methods S2 and Fig. S5). In MuSICA, the $\delta^{18}O$ of soil water is predicted based on liquid and vapour phase water isotope transport in the soil column and evaporative enrichment during soil evaporation. MuSICA allows two alternative formulations of the liquid water and water vapour effective diffusivities through the soil matrix. In the first formulation, these effective soil diffusivities increase linearly with the soil volumetric content of the liquid or vapour phase (Penman, 1940) while in the other formulation, proposed by

Moldrup et al. (2003), the influence of the pore-size distribution parameter and the total soil porosity is also taken into account. Here, we explore the consequences of using either the Penman or Moldrup soil diffusivity formulation on the prediction of the $\delta^{18}O$ of soil, xylem and leaf waters.

### 2.4.3 Canopy and gas exchange parameters

Grassland vegetation at the experimental site was parameterised in terms of canopy structure, the gas exchange properties of

leaves, as well as root distribution and hydraulic properties (Table S1). In theory, MuSICA could account for species mixtures and competition for water and light, but this would require parameters for every single species. As the mixed-species samples were dominated by *L. perenne* and *P. pratensis* with closely similar morphophysiology, we treated the vegetation sample as one plant type, described with one parameter set (Table S1).

The mean leaf area index (LAI; 2.6 ± 0.7) and the mean leaf zenithal angle (LZA; 58° ± 3°, corresponding to a leaf

inclination index (LII) close to zero, typical of a spherical leaf angle distribution) were estimated from compressed sward height measurements made throughout the 2005 to 2012 grazing seasons ($n = 74$ dates with a total of more than 7000 measurements) and calibration functions obtained from parallel measurements of compressed sward height, uncompressed sward height (estimated with a ruler), LAI and leaf zenithal angle (both determined with a LAI-2000, LI-COR, Nebraska, USA) at the site. The vertical distribution of leaf area in the canopy was described based on Wohlfahrt et al. (2003) (Fig. S6).

In the standard parameterisation, LAI and LII were set as constants, in agreement with the constant sward state imposed by management practices (see above). In the sensitivity analyses, we also tested the effect of observed variations of sward height, LAI and LII on modelled $\delta^{18}O$ of the different water compartments.



Leaf turnover is generally high in grassland (Chapman and Lemaire, 1996) including at our experimental site (Schleip et al., 2013). Thus, the co-dominant species (*L. perenne*, *P. pratensis*, *T. officinale* and *T. repens*) had a short and very similar mean leaf life span of ~460 growing degree days (GDD, with a base temperature of 4°C) throughout the vegetation period (Schleip et al., 2013). As leaf turnover is high, the photosynthetic characteristics of leaves were set constant in the standard

parameterisation. Leaf photosynthesis was modelled according to the Farquhar-von Caemmerer-Berry model (Farquhar et al., 1980). Values for the maximum rate of carboxylation ($V_{cmax}$), the light-saturated potential rate of electron transport ($J_{max}$) and other photosynthetic parameters were all taken from literature (Table S1). Leaf respiration rate was estimated from measurements made in the dark at the site (Ostler et al., unpublished) and was assumed to be partly inhibited during the day (e.g. Atkin et al., 1997).

Under well-watered conditions, stomatal conductance for water vapour ($g_s$) was simulated according to the Ball-Woodrow-Berry (BWB) model (Ball et al., 1987). This model has two parameters: $m_{gs}$, a species-specific non-dimensional parameter that determines the composite sensitivity of $g_s$ to net $CO_2$ assimilation and to relative humidity and $CO_2$ concentration at the leaf surface, and $g_0$, the basal (or minimal) stomatal conductance. Uncertainties exist regarding the slope parameter $m_{gs}$ and the intercept $g_0$ (Miner et al., 2017, and references therein). Values for $m_{gs}$ reported by Wohlfahrt et al. (1998) for 13

grassland species from differently managed sites ranged between 6.9 and 24.7, and values for the intercept $g_0$ (termed $g_{min}$ in their work) ranged between 12 and 193 mmol m$^{-2}$ s$^{-1}$. Likewise, a considerable range of nighttime stomatal conductance ($g_{night}$) has been reported for $C_3$ grasses: from 60 to 140 mmol m$^{-2}$ s$^{-1}$ (Ogle et al., 2012; Press et al., 1993; Snyder et al., 2003). Here, $g_{night}$ (together with leaf lamina water content $W$, see below) was manually adjusted by fitting MuSICA to diurnal measurements of leaf water δ$^{18}$O (Fig. S7). In the standard simulation, we used $m_{gs}$ = 10, a commonly used value for

$C_3$ vegetation (cf. Miner et al., 2017), $g_0$ = 10 mmol m$^{-2}$ s$^{-1}$ and $g_{night}$ = 30 mmol m$^{-2}$ s$^{-1}$. Finally, we tested the sensitivity of model predictions to variations of $m_{gs}$ and $g_0$ (see below).

The effect of edaphic drought on $g_s$ was considered by scaling $m_{gs}$ and $g_0$ with a function of predawn leaf water potential (Nikolov et al., 1995). This adds two extra model parameters whose values were sourced from the literature (Table S1) and results in a 50% reduction of $m_{gs}$ and $g_0$ at -1.5 MPa.

Characteristic dimensions of leaves and shoots for the calculation of boundary-layer conductance were estimated based on measurements on individual grass tillers. The width and length (0.1 and 7 cm, respectively) of the leaf blade of a 7 cm-tall grass tiller were taken as minimum and maximum values for the leaf dimensions, and the average leaf dimension was estimated as the square root of the area of such a leaf blade (0.8 cm). Values for minimum, maximum and average shoot dimensions were taken from sward height measurements (see above). The shelter factor was varied between 1 and 3.5

(Monteith and Unsworth, 1990), with very little consequences on the results. Parameter values for leaf optical properties, rain interception and wind attenuation were taken from the literature (Table S1).

In the model, total rooting depth was equated with topsoil depth (37 cm), as in Schnyder et al. (2006). The vertical distribution of fine roots in the soil column was assumed to follow a beta distribution with a maximum at 7 cm belowground (Fig. S8). The total amount of roots (g m$^{-2}$ of soil) was obtained from soil core sampling. The proportion of live roots was





derived from a 14-days long dynamic $^{13}CO_2/^{12}CO_2$ labelling experiment at the same site (Gamnitzer et al., 2009; Schleip, 2013; Ostler et al., 2016; Ostler et al., unpublished). Root mass data were converted to root lengths by assuming a specific root length of 100 m g$^{-1}$ (Picon-Cochard et al., 2012). Mean fine root radius was set to 0.15 mm (Picon-Cochard et al., 2012), and root xylem radial hydraulic resistance to 1.0 10$^{12}$ s m$^{-1}$.

### 2.4.4 Oxygen isotope composition of water pools

The steady-state $^{18}O$ enrichment of leaf water at the evaporative site ($\Delta^{18}O_{e,ss}$) was calculated according to (Farquhar and Lloyd, 1993; Farquhar and Cernusak, 2005):

$$\Delta^{18}O_{e,ss} = \alpha^+ (\alpha_k (1 - h) + h (\Delta^{18}O_v + 1)) - 1, \tag{2}$$

where $h$ is the air relative humidity, normalised at leaf temperature (estimated from the leaf energy budget), $\Delta^{18}O_v$ represents
the isotopic composition of atmospheric water vapour, expressed above that of xylem water, $\alpha^+$ is the isotope fractionation during liquid-vapour equilibrium at leaf temperature (Majoube, 1971) and $\alpha_k$ is the kinetic isotope fractionation during water vapour diffusion through stomata and leaf boundary layer. $\alpha_k$ was estimated at half-hourly time steps from stomatal and boundary-layer conductances for water vapour ($g_s$ and $g_b$):

$$\alpha_k = 1 + \frac{0.028/g_s + 0.019/g_b}{1/g_s + 1/g_b}, \tag{3}$$

Equation (3) uses the kinetic fractionation factor during molecular diffusion (28‰) reported by Merlivat (1978) and assumes laminar diffusion through the leaf boundary layer (Farquhar et al., 2007).

We modelled leaf water isotope enrichment at isotopic steady state ($\Delta^{18}O_{leaf,ss}$) using the two approaches introduced earlier. In the "two-pool" simulation, we used a constant value for φ of 0.39, which was chosen such that the observed $\Delta^{18}O_{leaf}$ was on average predicted without bias. In the sensitivity analysis, φ was varied between -0.20 and 0.50 based on the range of
values reported previously for a variety of grass species (Helliker and Ehleringer, 2000; Gan et al., 2003; see Discussion). In the "Péclet" simulation, $\Delta^{18}O_{leaf,ss}$ was related to $\Delta^{18}O_{e,ss}$ using the Péclet number, as described by Farquhar and Lloyd (1993):

$$\Delta^{18}O_{leaf,ss} = \Delta^{18}O_{e,ss} \frac{1 - e^{-p}}{p}, \tag{4}$$

with $p$ the Péclet number. The latter is calculated as $p = EL/(CD)$ where $L$ (m) is the effective path length, $E$ (mol m$^{-2}$ s$^{-1}$) is
the leaf transpiration rate, $C = 55500$ mol m$^{-3}$ is the molar density of liquid water and $D$ (m$^2$ s$^{-1}$) the diffusivity of $H_2^{18}O$ in liquid water (Farquhar and Lloyd, 1993; Cuntz et al., 2007). In line with the original notion of the Péclet model, one single value of $L$ was applied to the dataset, which was again adjusted such that $\Delta^{18}O_{leaf}$ was predicted without bias.

Two supplementary experiments were also conducted to directly test the relevance of the Péclet effect in the co-dominant pasture species *L. perenne* and *D. glomerata*. These are described in the Supplement.





As leaf water is not in isotopic steady state for extended periods of the day (Fig. S9), an equation for non-steady state enrichment of leaf water was used in addition to Eq. (2)-(4). Using isotopic mass balance of leaf water and assuming that Eq. (4) holds true also in the non-steady state (Farquhar and Cernusak, 2005), the time evolution of $\Delta^{18}O_{leaf}$ was modelled as (see also Farquhar et al., 2007):

$$\frac{d\left(W \Delta^{18}O_{leaf}\right)}{dt} = -\frac{E}{\alpha_k \alpha^+ (1-h)} \frac{p}{1-e^{-p}} (\Delta^{18}O_{leaf} - \Delta^{18}O_{leaf,ss}), \qquad (5)$$

where $W$ (mol m$^{-2}$) denotes leaf lamina water content, expressed on a leaf area basis.

A 27-h time series of community-scale $\delta^{18}O_{leaf}$ observed at the site in August 2005 (Fig. S7) was used to fine-tune the parameters controlling leaf water enrichment in MuSICA (mesophyll water content and night-time and minimum stomatal conductance) within the range of values expected for temperate grasslands (for parameter values see Table S1). Because
MuSICA predicts different leaf-level variables (e.g. $g_s$, $g_b$, $h$, $E$, $\Delta^{18}O_{leaf,ss}$,…) for sunlit, shaded, wet or dry leaves at different levels within the canopy, assimilation-weighted canopy averages of $\delta^{18}O_{leaf}$ and $\Delta^{18}O_{leaf}$ were first calculated at every time step before performing comparisons with observed data.

### 2.5 Sensitivity analysis

A sensitivity analysis was conducted in order to quantify the responsiveness of predicted midday $\delta^{18}O$ of leaf, stem and soil
water to plant morpho-physiological parameters that were expected to affect those predictions based on theoretical considerations and/or observed parameter variation at the site. As the leaf water enrichment submodels are embedded in the process-based model MuSICA, the effect of parameters not included in the leaf water $\delta^{18}O$ models per se could be evaluated. Based on the ceteris paribus principle, the sensitivity was tested by varying one parameter while keeping all other parameters the same as in the standard MuSICA parameter set (Table S1). For a sensitivity run, the parameter was not decoupled from
the equations in MuSICA, hence changing one parameter value at the same time affected all equations containing this parameter and all dependent variables. Parameter effects (sensitivities) were quantified by calculating the mean differences from the reference run as $(\sum_{i=1}^{n} (\delta_{sens,i} - \delta_{ref,i}))/n$, with $\delta_{sens,i}$ the $\delta^{18}O$ of a given water compartment (leaf, stem, or soil at 7 or 20 cm depth) in a sensitivity run and $\delta_{ref,i}$ that in the reference run, for a day $i$. Besides, the standard deviations of the differences between $\delta_{sens,i}$ and $\delta_{ref,i}$ were calculated for each parameter and water compartment, which illustrate how strongly
the effect of that parameter differs from day to day, and hence how strongly it depends on the instantaneous conditions encountered on one specific day.

The high and low parameter values for the sensitivity analyses were chosen according to the range observed for grasses or grassland species, as reported in the literature or observed at the site (see Supplement). Values for individual parameters of the sensitivity analysis were set at –0.20 and 0.50 for $\varphi$, 1 or 12 mol m$^{-2}$ for leaf lamina water volume ($W$), 7 or 25 for the
slope of the BWB model ($m_{gs}$), 0 or 193 mmol m$^{-2}$ s$^{-1}$ for the intercept of the BWB model ($g_0$), 0.6 or 3.8 m$^2$ m$^{-2}$ for leaf area index (LAI), 3.6 or 11.7 cm for canopy height ($h_{canopy}$), 20 or 140 µmol m$^{-2}$ s$^{-1}$ for the maximum rate of carboxylation at 25$^o$C



($V_{cmax}$) and 32 or 224 µmol m$^{-2}$ s$^{-1}$ for potential rate of electron transport at 25°C ($J_{max}$) and 0.08 or 0.265 m for the mean of the vertical root distribution ($\mu_{root}$). $V_{cmax}$ and $J_{max}$ were altered *in tandem* in order to keep the ratio $J_{max}/V_{cmax}$ at 1.6. Apart from those plant morpho-physiological parameters, the effect of alternative submodels for the liquid and vapour effective diffusivity in the soil was tested by replacing the Moldrup formulation by the Penman one. In addition, we used IsoGSM-predicted $\delta^{18}O_{rain}$ and $\delta^{18}O_{vapour}$ data instead of measured $\delta^{18}O_{rain}$ and $\delta^{18}O_{vapour}$ data for the isoforcing of MuSICA in order to illustrate the use of having local rainwater $\delta^{18}O$ data.

**2.6 Statistics**

For comparison of predicted and observed data, we calculated the mean bias error (MBE $= \overline{P} - \overline{O}$, where $\overline{P}$ is the mean predicted value and $\overline{O}$ the mean observed value) between observed and predicted $\delta^{18}O$ (or $\Delta^{18}O$), and the mean absolute error (MAE $= (\sum_{i=1}^{n} |P_i - O_i|)/n$), where $P_i$ is the predicted and $O_i$ is the observed value at time $i$, and $n$ is the number of values; Willmott and Matsuura, 2005).

Data analyses were performed in R, version 3.4.2 (R Core Team, 2017) and RStudio, version 1.1.383 (RStudio Team, 2016).

**3 Results**

**3.1 Rainfall, $\delta^{18}O$ of precipitation and vapour**

Growing season rainfall amounts and distribution differed between years, with total precipitation in the main growing period (May to August) varying between 321 mm (2006) and 514 mm (2010) (Fig. 1a). The mean $\delta^{18}O_{rain}$ signal tended to increase in the first half of the vegetation period and decrease later in the season (Fig. 1b). However, individual rain events sometimes differed markedly from the mean pattern, with excursions of up to +4.5‰ and -6.2‰ relative to the mean of the same month (Fig. 1b). The $\delta^{18}O_{vapour}$ signal followed similar mean trends (Fig. 1c), and exhibited a significant correlation ($P < 0.001$) with the $\delta^{18}O$ of the previous rain event.

**3.2 Soil water**

Volumetric soil water content (SWC) predicted by MuSICA using the standard parameterisation (Table S1) exhibited strong seasonal and inter-annual variations. With SWC values (in m$^3$ m$^{-3}$) expected to vary between 0.19 (permanent wilting point) and 0.46 (field capacity), a SWC of less than 0.25 at 7 cm belowground corresponds to <25% of the maximum plant available water at this depth, and is therefore a good indicator of edaphic drought. Each year, soil moisture at 7 cm fell below this threshold, but with a timing that differed from one year to the next (Fig. 1d).

The observed $\delta^{18}O_{soil}$ was generally more enriched at 7 cm than at 20 cm belowground (Table 1; Fig. 2a, b). This relative enrichment with shallower depth was particularly large in the first half of the vegetation period, and averaged 1.7‰ in the





entire data set. The total observed range of $\delta^{18}O_{soil}$ differed somewhat between the two depths and was 7.8‰ at 7 cm, i.e., 16% greater than at 20 cm (Table 1).

In most years, $\delta^{18}O_{soil}$ followed the rain pattern and increased during the course of the vegetation period at both depths (Fig. 2a, b). This increase was generally more pronounced at 7 cm than at 20 cm. Overall, the seasonal patterns of $\delta^{18}O_{soil}$

were quite dynamic, with considerable differences between individual years.

MuSICA simulations with the standard parameterisation (Table S1) predicted the multi-seasonal dynamics of $\delta^{18}O_{soil}$ well (Fig. 2a, b) except in 2006 when local data of $\delta^{18}O_{rain}$ were not available for the iso-forcing (Fig. 1b) and $\delta^{18}O_{rain}$ data were taken from the global atmospheric model IsoGSM, once corrected for the mean model-data offset (Figs. S2–S4). The seasonal trends and monthly fluctuations of observed $\delta^{18}O_{soil}$ were reproduced with relatively small error (MAE of 1.1‰ and

0.8‰ at 7 and 20 cm, respectively). Also, the bias was small as MuSICA overestimated $\delta^{18}O_{soil}$ by 0.8‰ and 0.5‰ at 7 and 20 cm, respectively.

### 3.3 Stem water

Observed $\delta^{18}O_{stem}$ generally matched and followed that of $\delta^{18}O_{soil}$ at 7 cm, independently of SWC, season and year (Figs. 2b, c, 3a and S10). A similarly close relationship did not exist between $\delta^{18}O_{stem}$ and $\delta^{18}O_{soil}$ at 20 cm (Fig. 3c). Thus,

the MAE (0.7‰) between $\delta^{18}O_{stem}$ and $\delta^{18}O_{soil}$ at 7 cm depth was about three times smaller than that for $\delta^{18}O_{stem}$ and $\delta^{18}O_{soil}$ at 20 cm, and only slightly greater than the MAE of 0.5‰ between the replicate samples of soil water taken at 7 cm (Table 2). Remarkably, for 90% of all days on which the soil was classified as 'dry' (predicted SWC<0.25), $\delta^{18}O_{stem}$ was still closer to $\delta^{18}O_{soil}$ at 7 cm than to $\delta^{18}O_{soil}$ at 20 cm, indicating that root uptake did not shift to the lower part of the profile during edaphic drought.

Barnard et al. (2006) showed that the $\delta^{18}O$ of (pseudo-)stem water in grasses is very close to that of the water taken up by the root systems of grasses (see also Liu et al., 2017), meaning that root water uptake operates without $^{18}O$ isotope fractionation. MuSICA simulations were based on this assumption and reproduced very similar relationships between $\delta^{18}O_{stem}$ and $\delta^{18}O_{soil}$ as those observed at both depths, with very similar coefficients of determination (Figs. 2-3). Importantly, the close correspondence of $\delta^{18}O_{stem}$ with $\delta^{18}O_{soil}$ at 7 cm depth was not affected by changes in SWC predicted by MuSICA (Fig. 3).

Again, the strongest disagreement between predicted and observed $\delta^{18}O_{stem}$ occurred in 2006 (Fig. 2c), when observations of local $\delta^{18}O_{rain}$ were unavailable.

### 3.4 Leaf water

Midday leaf water $\delta^{18}O$ ($\delta^{18}O_{leaf}$) exhibited by far the greatest observed $\delta^{18}O$ variations in the entire dataset (Table 1). Also, $\delta^{18}O_{leaf}$ was unique in that it did not exhibit a general trend during the vegetation period ($P = 0.5$; right panel in Fig. 2d). This

implied that the observed large temporal variation of $\delta^{18}O_{leaf}$ was the result of a short-term response. Because, on average, $\delta^{18}O_{stem}$ increased over the vegetation period while $\delta^{18}O_{leaf}$ did not, $\Delta^{18}O_{leaf}$ exhibited a significant decreasing trend over the vegetation period, with a decrease of 0.5‰ per month ($P = 0.01$; right panel in Fig. 2e), most likely driven by an increase of





relative humidity over the growing season (not shown). Conspicuous short-term, parallel increases/anomalies of $\delta^{18}O_{leaf}$ and $\Delta^{18}O_{leaf}$ (i.e. changes of $\delta^{18}O_{leaf}$ largely independent of variations of $\delta^{18}O_{stem}$) occurred occasionally in different years, e.g. in spring of 2008, late spring and early fall of 2009, and early summer of 2010.

Predictions of $\Delta^{18}O_{leaf}$ with MuSICA agreed best with observations using the two-pool model with $\varphi = 0.39$ ($R^2 = 0.42$;

Table 2) in the standard MuSICA parameterisation. This result was robust for different soil water conditions. Unbiased predictions of $\Delta^{18}O_{leaf}$ were best obtained by decreasing $\varphi$ by 0.03 (i.e. setting $\varphi$ to 0.36) under dry soil conditions (SWC < 0.25) and increasing it by 0.01 (i.e. setting $\varphi$ to 0.40) under moist soil conditions (SWC ≥ 0.25), but this was an insignificant adjustment that did not change the overall coefficient of determination between observed and predicted $\Delta^{18}O_{leaf}$. The agreement between observed and predicted $\Delta^{18}O_{leaf}$ was always weaker when using the Péclet model. Fixing the

effective path length ($L$) at a certain value led to predictions that were systematically biased for either dry or moist soil conditions (Table 3). Unbiased predictions of $\Delta^{18}O_{leaf}$ in conditions of different SWC were only obtained when increasing $L$ (from 0.162 m to 0.235 m) for dry soil conditions and decreasing $L$ for moist soil conditions (from 0.162 m to 0.142 m).

MuSICA predictions of $\delta^{18}O_{leaf}$ and $\Delta^{18}O_{leaf}$ obtained with the standard parameterisation agreed well with observations at all time scales (Figs. 2d, e, S7 and S9), with low or no bias (MBE of 0.3‰ and 0.0‰, respectively) and an MAE for $\delta^{18}O_{leaf}$ of

1.6‰, i.e., 10% of the total variations of $\delta^{18}O_{leaf}$ in the entire dataset (Tables 1-2). The superiority of the two-pool model compared to the Péclet model for predicting $\Delta^{18}O_{leaf}$ in our dataset was underlined by the absence of a relation between transpiration rate and the proportional difference between the observed $\Delta^{18}O_{leaf}$ and $\Delta^{18}O$ predicted by the Craig-Gordon model (Fig. S11).

### 3.5 Relationships between soil and atmosphere water status, transpiration, canopy conductance and $^{18}O$ enrichment
### of bulk leaf water

Multiple regression analysis demonstrated significant effects of air relative humidity ($P < 0.01$) and SWC ($P < 0.05$) on both observed and predicted $\Delta^{18}O_{leaf}$ (Table 4). $\Delta^{18}O_{leaf}$ increased with decreasing air relative humidity and SWC (Figs. 4a, b and 5a, b). The analysis also indicated a weakly significant interaction of air relative humidity and SWC effects on both observed ($P = 0.080$) and predicted ($P = 0.073$) $\Delta^{18}O_{leaf}$ (Table 4). The dependence of transpiration (estimated with MuSICA) on air

VPD was strongly modified by SWC (Fig. 4c), with high air VPD consistently driving high transpiration rates only under wet soil conditions (SWC ≥ 0.25). Accordingly, the effect of dry soil conditions on $\Delta^{18}O_{leaf}$ was most evident at low air humidity (Figs. 4a, b and 5a, b) and was connected with a decrease of canopy conductance ($g_{canopy}$) (Fig. 5c), estimated here as the ratio of ecosystem-scale transpiration rate and air VPD.

### 3.6 Sensitivity analysis

Increasing (decreasing) the proportion of un-enriched leaf water ($\varphi$) and leaf lamina water volume ($W$) led to a strong reduction (increase) in $\delta^{18}O_{leaf}$ (Figs. 6a, b). These changes in leaf-level parameters had no effect on $\delta^{18}O_{soil}$ or $\delta^{18}O_{stem}$. Alterations of stomatal responsiveness ($m_{gs}$), minimum conductance ($g_0$), maximum carboxylation ($V_{cmax}$) or electron



transport ($J_{max}$) rates and LAI had similar directional effects on predicted $\delta^{18}$O of soil, stem and leaf water but the strength of the effects differed for the different ecosystem water pools (Fig. 6). Stronger effects were found on $\delta^{18}$O$_{leaf}$ and $\delta^{18}$O$_{soil}$ at 20 cm, compared to $\delta^{18}$O$_{stem}$ or $\delta^{18}$O$_{soil}$ at 7 cm that tended to vary in close harmony. Generally, a change of the parameter value caused an opposite change of the predicted $\delta^{18}$O of a given pool. The (unanticipated) sensitivity of $\delta^{18}$O$_{soil}$ to plant

morpho-physiological parameters was mediated by the effect of those parameters on plant transpiration rate (not shown), which in turn altered the residence time of soil water at the lower depth. For example, lower $V_{cmax}$ and $J_{max}$ values, not accompanied by a change in stomatal responsiveness $m_{gs}$, led to a decrease in transpiration rate which caused an increase in the percolation of growing season rain water to the lower part of the soil profile (Figs. 7a and 8). In comparison, the $^{18}$O-depleted (winter) signal persisted longer in the lower profile at intermediate (Fig. 7b) or high (Fig. 7c) $V_{cmax}$ and $J_{max}$, as

higher transpiration rates during the growing season inhibited the replenishment of deep soil layers with summer rainfall.

Apart from LAI, other shoot characteristics, such as canopy height (Fig. 6f), leaf inclination, shoot shelter factor, leaf size and shoot size (not shown) had a very small or no effect on predicted $\delta^{18}$O$_{leaf}$, $\delta^{18}$O$_{stem}$ and $\delta^{18}$O$_{soil}$.

The formulation of the water vapour diffusivity through the soil matrix (Fig. 6i) and the average rooting depth (Fig. 6h) affected $\delta^{18}$O$_{soil}$ (and more strongly so at the lower depth), while the effect on $\delta^{18}$O$_{stem}$ and $\delta^{18}$O$_{leaf}$ was much weaker. Not

accounting for the pore-size soil particle distribution parameter in the soil diffusivity formulation caused a greater overestimation of $\delta^{18}$O$_{soil}$, especially at 20 cm belowground where the MBE reached 1.3‰, compared to 0.5‰ in the standard run. Shifting the root distribution closer to the soil surface had little effect on $\delta^{18}$O$_{soil}$ at both depths. Conversely, shifting it towards greater depth (Fig. S8) led to an overestimation of $\delta^{18}$O$_{soil}$, especially at 20 cm (Fig. 6h), and increased MAE in the relationship between $\delta^{18}$O$_{stem}$ and $\delta^{18}$O$_{soil}$ at both soil depths (not shown).

We also tested the effect of the choice of the water isotope forcing of MuSICA ($\delta^{18}$O$_{rain}$ and $\delta^{18}$O$_{vapour}$). In general, the agreement between predicted and observed ecosystem water pool $\delta^{18}$O was much better when MuSICA was forced using locally measured $\delta^{18}$O$_{rain}$ and $\delta^{18}$O$_{vapour}$ data (Fig. 6j). The MBE for the $\delta^{18}$O of the different water pools was 3.1 to 6.7-fold greater when using the IsoGSM-based isotope forcing, and the MAE was 1.5 to 2.6-fold higher.

## 4 Discussion

### 4.1 Model realism

An isotope-enabled, process-based soil-plant-atmosphere model, MuSICA, generated realistic predictions of multi-seasonal dynamics of $\delta^{18}$O in soil, pseudo-stem (xylem) and midday leaf water, as well as of the $^{18}$O enrichment of leaf water in a drought-prone temperate grassland ecosystem. Throughout the vegetation periods of seven consecutive years (1) model bias (MBE) was low, (2) the range of $\delta^{18}$O variations of the different ecosystem water pools was similar in the predictions and

observations, and (3) prediction error (MAE) was less than 15% of the total observed range of $\delta^{18}$O in the different ecosystem water pools and about twice the size of the MAE for the duplicate samples of the different pools. The relationships between observed $\Delta^{18}$O$_{leaf}$ and variables related to the water cycle such as SWC, air relative humidity,





transpiration and canopy conductance were well captured by the model. Although MuSICA is a detailed and locally-parameterised model, this general agreement between model predictions and observations is remarkable given that model parameters describing the relevant physical features or functional relationships of soil and vegetation were held constant with one single value for the entire mixed-species ecosystem. This is a striking outcome given that predicted $\delta^{18}O$ were found to

be quite sensitive to several (but not all) plant morpho-physiological parameters (Fig. 6). The greater scatter in the observed relationship between $\Delta^{18}O_{leaf}$ and relative humidity compared to predictions (Fig. 4) probably resulted largely from sampling effects, in addition to analytical error. Such sampling effects could include small-scale spatial variation of soil properties, or spatio-temporal variation of LAI, nutrient levels and root distribution, a regular feature of grazed grassland (e.g. Schnyder et al., 2006, 2010). Also, Webb and Longstaffe (2003) observed differences of several per mil in $\delta^{18}O_{soil}$ in the top 5 cm over

distances of about 10 m in a sand dune grassland. Such spatial variations would inherently cause greater scatter in the observations compared to the model predictions.

Prediction of $\delta^{18}O_{stem}$ at a given point in time is a real challenge, as $\delta^{18}O_{stem}$ is influenced by numerous factors, including the temporal distribution of rainfall amounts and its associated isotopic composition, transport and mixing of rainwater with soil water, the depth distribution of root water uptake in the soil and soil evaporation. These processes are described explicitly in

MuSICA. Importantly, a better agreement between predicted and observed $\delta^{18}O_{soil}$ at 7 cm and $\delta^{18}O_{stem}$ was obtained when the $\delta^{18}O$ of meteoric water was taken from local measurements rather than given by the isotope-enabled atmospheric model IsoGSM (Fig. 6j). This result is not surprising given the significant spatial and temporal variation of rainfall at weekly and sub-kilometre scales (Fiener and Auerswald, 2009) and the comparatively large grid size of the IsoGSM model simulations (*ca*. 200 km × 200 km). Our model sensitivity analysis also revealed a better predictive power of the soil diffusivity

formulation proposed by Moldrup et al. (2003) over that proposed by Penman (1940) to reproduce the observed isotopic composition of all the ecosystem water pools (Fig. 6i). This superiority was likely related to the effect of accounting for the soil pore size distribution parameter for describing the effective liquid water and water vapour diffusivity through the soil matrix and estimating this parameter from the soil water retention curve parameters measured at the site.

## 4.2 Xylem water originates from shallow soil depths independently of season and soil water content

The comparison of observed $\delta^{18}O_{stem}$ and $\delta^{18}O_{soil}$ (Fig. 3a) strongly suggested that root water uptake occurred at very shallow depths throughout the vegetation periods and independently of changes in SWC. This was well supported by the standard MuSICA runs (Fig. 3b) and sensitivity analysis (Fig. 6h) showing that $\delta^{18}O_{stem}$ was well predicted by the model only when roots were distributed in very shallow horizons, a feature typical of intensively grazed grasslands (Troughton, 1957; Klapp, 1971). These results are in line with a recent study of Prechsl et al. (2015) that did not find an increasingly deeper root water

uptake upon soil drying in an alpine and a lowland grassland system in Switzerland. Also, grasses continued to rely on water in the uppermost soil layer during soil water scarcity in a mesic Savanna in South Africa, in which $C_4$ grasses were growing together with saplings and trees (Kulmatiski and Beard, 2013), and in a tallgrass prairie in the US dominated by $C_4$ grasses and $C_3$ shrubs and forbs (Nippert and Knapp, 2007a, b). In the present case, the shallow rooting depth may have been



exacerbated by the high grazing pressure and consequent limitations in resource allocation to root growth. Besides, root water uptake of the co-dominant species presumably did not shift to lower horizons due to relatively low water contents in the lower part of the soil profile, as predicted by MuSICA (Fig. S12).

Predictions of $\delta^{18}O_{soil}$, particularly at 20 cm, were influenced markedly by estimates of LAI and by changes of $V_{cmax}$, $J_{max}$,

and stomatal conductance responsiveness ($m_{gs}$) or minimal value ($g_0$). This resulted from the effect of those parameters on total canopy transpiration, that in turn altered the dynamics of soil water and hence of the mixing of $^{18}O$-depleted winter and $^{18}O$-enriched summer precipitation with soil water at different depths. For instance, an increase in transpiration rate caused by a high $m_{gs}$ led to a decrease in $\delta^{18}O_{soil}$ at 20 cm during the course of the growing season and a growing divergence between observations and predictions, particularly in years with low growing season precipitation (data not shown). This was

likely caused by the fact that $^{18}O$-enriched summer rain mainly recharged the upper soil layer in this scenario (as this had been desiccated extensively because of the higher transpiration resulting from the higher $m_{gs}$). So, summer rains would contribute less to wetting of the lower profile. Conversely, if $m_{gs}$ was set to a low value, predicted $\delta^{18}O_{soil}$ at 20 cm increased throughout the vegetation period. According to the same mechanism, the effect of $m_{gs}$ on $\delta^{18}O_{soil}$ was negligible when growing season rainfall was high in 2010. The effects of changing $V_{cmax}$ and $J_{max}$, LAI and minimum conductance on

predicted $\delta^{18}O_{soil}$ at 20 cm were very similar to $m_{gs}$, suggesting that these parameters acted *via* the same mechanism, that is canopy conductance for water vapour that is controlled largely by the (integrated) stomatal conductance of all leaves within the canopy. Thus, the effect of $V_{cmax}$ and $J_{max}$ was likely indirect, resulting from altered assimilation rates impacting stomatal conductance.

## 4.3 Evidence for a two-pool model of leaf water $^{18}O$ enrichment

The $\Delta^{18}O_{leaf}$ data were well predicted with a two-pool model and a constant fraction of un-enriched water in bulk leaf water ($\varphi \approx 0.39$). This model was valid for a wide range of atmospheric and soil water conditions in seven consecutive growing seasons. Inclusion of a Péclet effect reduced the closeness of fit between measured and modelled $\Delta^{18}O_{leaf}$ under all environmental conditions. Supplementary studies also supported the two-pool model for pasture species. There was no relationship between the proportional difference between measured leaf water enrichment and that predicted by the Craig-

Gordon model ($1 - \Delta^{18}O_{leaf}/ \Delta^{18}O_{e,ss}$) and transpiration rate in either *L. perenne* plants grown in a controlled environment at different relative humidities and water availabilities, or *D. glomerata* leaves measured using an online transpiration isotope method (Figs. S13-14). A two-pool model was also suggested by the diurnal time courses of $\delta^{18}O_{leaf}$ in this grassland (Fig. S7) and in a broadleaf and a coniferous tree species (Bögelein et al., 2017). When interpreted with the Péclet model, the two-pool model implies a constant Péclet number and inverse variation of transpiration rate and effective path length ($L$).

Dynamic changes of $L$ in response to varying transpiration have been noted before, mainly in controlled conditions, and interpreted in terms of changing contributions of different paths (symplastic, apoplastic, and transcellular) of water movements to the stomatal pore (Barbour and Farquhar, 2003; Kahmen et al., 2008; Song et al., 2013; Loucos et al., 2015;





Cernusak et al., 2016). Increases of *L* in response to drought, as found in this work, have also been observed previously in *Vitis vinifera* by Ferrio et al. (2012), and were connected with variations in leaf lamina hydraulic conductance.

In principle, failure to detect a Péclet effect could be related to the presence of major veins and associated ground tissue of the grass leaves (Holloway-Phillips et al., 2016) or errors associated with non-steady-state effects on $^{18}$O enrichment of bulk

leaf water (Cernusak et al., 2016). However, MuSICA predictions of $\Delta^{18}O_{leaf}$ did account for non-steady state effects and were generally consistent with observed $\Delta^{18}O_{leaf}$. The φ value used in our simulations is in the upper range of φ values reported for grasses. Liu et al. (2017) observed species-specific φ values ranging from -0.05 to 0.43 in two $C_3$ and three $C_4$ grasses, with no obvious effect of vapour pressure deficit on φ. Gan et al. (2003) presented φ values between *ca.* 0.16 and 0.41 in maize, with lower values coming from leaves with the mid-vein removed. Considering a similar effect of vein

removal would move our observed φ to about 0.2, i.e., close to the mean φ value reported by Cernusak et al. (2016) for a wide range of (non-grass) species.

## 4.4 Atmospheric and edaphic effects on the $^{18}$O enrichment of leaf water

The strong response of $\Delta^{18}O_{leaf}$ to air relative humidity has been observed and discussed previously (e.g. Farquhar et al., 2007; Cernusak et al., 2016), in addition to soil moisture (Ferrio et al., 2012). We are not aware of a previous study that

disentangled the separate effects of atmospheric and soil humidity on $\Delta^{18}O_{leaf}$, either in field or controlled conditions. Notably, the responses observed in our work were corroborated by theoretical predictions as implemented in MuSICA. Modelled transpiration rate and stomatal conductance were greatly reduced under dry soil conditions, leading to higher kinetic fractionation $\alpha_k$ (Eq. 3) but lower $\alpha^+$ (Majoube, 1971) and relative humidity *h*, because of the warmer leaf temperatures. The net effect was a greater $\Delta^{18}O_{leaf}$ predicted by MuSICA under dry soil conditions, in agreement with

observations. This demonstrated that other vegetation parameters that affected the $^{18}$O-enrichment in our sensitivity analysis (e.g. the un-enriched fraction φ or the effective mixing length *L*, mesophyll water content *W* or LAI) but were not considered drought-sensitive, did not seem the main drivers of the enhancement of $\Delta^{18}O_{leaf}$ during edaphic drought.

## 5 Conclusions

This work highlights the usefulness of mechanistic $^{18}$O-enabled modelling for explorations and quantitative analyses of the

ecohydrology of ecosystems. Such modelling demonstrated here that (1) a single set of plant parameters and site-specific soil properties was enough to capture the main $\delta^{18}O$ dynamics of ecosystem water pools, despite the species mixture characteristic of grassland ecosystems, (2) water uptake occurred from shallow soil depths throughout dry and wet periods in all years, as confirmed by soil and xylem $\delta^{18}O$ data and model sensitivity analysis on mean rooting depth and (3) $\Delta^{18}O_{leaf}$ responded to both soil and atmospheric moisture, and was best described when leaf water was separated into two non-mixing

water pools, a result that could be captured solely based on the drought sensitivity of leaf stomatal conductance and photosynthetic capacity, and resulting effects on the leaf energy balance. Demonstration of an effect of soil drying on





$\Delta^{18}O_{leaf}$ together with reduced stomatal conductance is of great interest for retrospective studies of the functional components controlling water use efficiency of plants. If imprinted on the $\delta^{18}O$ of plant cellulose, such an effect may also help identify drought events in archived materials, such as the grassland vegetation samples of the Park Grass experiment or herbaria.

*Supplement*

The supplement related to this article is available online.

*Author contribution*

JO, RTH and HS designed the study. RTH analysed the data and performed the modelling with guidance by JO. IS and UO designed the sampling scheme and set up, tested the water extraction unit and performed the diurnal water sampling. RS performed the isotope analysis. SHV analysed the eddy flux data. MMB performed the supplementary controlled environment experiments. RTH and HS wrote the paper. All authors contributed to the discussion and revision.

*Competing interests*

The authors declare that they have no conflict of interest.

*Acknowledgements*

This research was supported by the Deutsche Forschungsgemeinschaft (SCHN 557/9-1), the Agence Nationale de la
Recherche (ANR-13-BS06-0005) and European Union's Seventh Framework Programme (FP7/2007-2013) (grant agreement No. 338264). We thank Erna Eschenbach†, Angela Ernst-Schwärzli, Anja Schmidt, Monika Michler, Hans Vogl, Richard Wenzel and Lenka Plavcová for technical assistance, Kei Yoshimura for sharing the IsoGSM $\delta^{18}O_{rain}$ and $\delta^{18}O_{vapour}$ data, Wolfgang Durner and Alina Miller for providing soil data, and Iris Köhler for previous discussion.

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





**Table 1: Minimum, maximum, mean, and range for the observed $\delta^{18}O$ of grassland ecosystem water pools (soil water at 20 and 7 cm depth, and stem and bulk leaf water) and $^{18}O$-enrichment of leaf water ($\Delta^{18}O$). Samples were collected at approximately biweekly intervals during the vegetation periods of 2006-2012.**

| | $\delta^{18}O$ (‰) | | | |
| --- | --- | --- | --- | --- |
| | Min | Max | Mean | Range |
| Soil water at 20 cm | -12.3 | -5.6 | -8.4 | 6.7 |
| Soil water at 7 cm | -11.3 | -3.5 | -6.7 | 7.8 |
| Stem water | -10.4 | -3.3 | -6.5 | 7.1 |
| Leaf water | -3.5 | 12.0 | 4.1 | 15.5 |
| | $\Delta^{18}O$ (‰) | | | |
| Leaf water | 4.7 | 18.2 | 10.5 | 13.5 |

**Table 2: $R^2$, mean bias error (MBE) and mean absolute error (MAE) for the comparison between predicted and observed $\delta^{18}O_{leaf}$, $\delta^{18}O_{stem}$, and $\delta^{18}O_{soil}$ at 7 cm ($\delta^{18}O_{soil\ 7}$) or 20 cm depth ($\delta^{18}O_{soil\ 20}$). Predictions were made with the standard MuSICA parameterisation given in Table S1. Values in parentheses exclude the data from year 2006. The last column presents the MAE**

10 **between the replicate samples collected on the different dates. MBE and MAE values are given in ‰.**

| | $R^2$ | MBE | MAE | MAE obs/obs |
| --- | --- | --- | --- | --- |
| $\delta^{18}O_{soil\ 20}$ | 0.79 (0.79) | 0.5 (0.6) | 0.8 (0.8) | 0.6 (0.5) |
| $\delta^{18}O_{soil\ 7}$ | 0.56 (0.72) | 0.8 (0.5) | 1.1 (0.9) | 0.5 (0.5) |
| $\delta^{18}O_{stem}$ | 0.46 (0.60) | 0.4 (0.2) | 1.1 (0.9) | 0.4 (0.4) |
| $\delta^{18}O_{leaf}$ | 0.43 (0.43) | 0.3 (0.2) | 1.6 (1.5) | 0.8 (0.7) |





**Table 3:** $R^2$, **mean bias error (MBE) and mean absolute error (MAE) for the comparison between predicted and observed** $\Delta^{18}O_{leaf}$ **obtained with different values of the proportion of unenriched leaf water (φ) in the two-pool model, or effective path lengths (*L*) in the Péclet model for the prediction of** $\Delta^{18}O_{leaf}$. **Best predictions are highlighted in bold print. The agreement between predictions and observations was tested for the entire data set (*n* = 83), or the moist (SWC ≥0.25; *n* = 57) or dry soil subsets (SWC <0.25; *n* = 26). The standard MuSICA parameterisation used a constant φ = 0.39 for all conditions in all years. MBE and MAE values are given in ‰.**

| Model | SWC | $R^2$ | MBE | MAE |
|---|---|---|---|---|
| Two-pool | | | | |
| φ = 0.36 | all | 0.42 | 0.5 | 1.5 |
| | moist | 0.48 | 0.7 | 1.2 |
| | **dry** | **0.38** | **0.0** | **2.2** |
| φ = 0.39 | **all** | **0.42** | **0.0** | **1.4** |
| | moist | 0.48 | 0.2 | 1.0 |
| | dry | 0.38 | −0.6 | 2.2 |
| φ = 0.40 | all | 0.42 | −0.3 | 1.4 |
| | **moist** | **0.48** | **0.0** | **1.0** |
| | dry | 0.38 | −0.8 | 2.3 |
| Péclet | | | | |
| *L* = 0.142 m | all | 0.24 | 0.5 | 1.9 |
| | moist | 0.36 | 0.0 | 1.1 |
| | dry | 0.12 | 1.8 | 3.5 |
| *L* = 0.162 m | all | 0.21 | 0.0 | 2.0 |
| | moist | 0.33 | −0.6 | 1.2 |
| | dry | 0.10 | 1.3 | 3.6 |
| *L* = 0.235 m | all | 0.15 | −1.6 | 2.9 |
| | moist | 0.26 | −2.3 | 2.4 |
| | dry | 0.05 | 0.0 | 3.9 |





**Table 4: Results of a multiple regression analysis of the effects of relative humidity (RH) and soil water content (SWC) on $^{18}$O-enrichment of leaf water as observed and as predicted by MuSICA with standard parameterisation. SE, standard error; *P*, significance level.**

| Parameter | observed | | | predicted | | |
|---|---|---|---|---|---|---|
| | Estimate | SE | *P* | Estimate | SE | *P* |
| RH | −0.31 | 0.09 | 0.001 | −0.29 | 0.06 | <0.001 |
| SWC | −41.4 | 19.2 | 0.034 | −25.2 | 11.4 | 0.030 |
| RH × SWC | 0.59 | 0.34 | 0.080 | 0.36 | 0.20 | 0.073 |
| Regression model | $R^2$ | | | $R^2$ | | |
| | 0.44 | | | 0.74 | | |

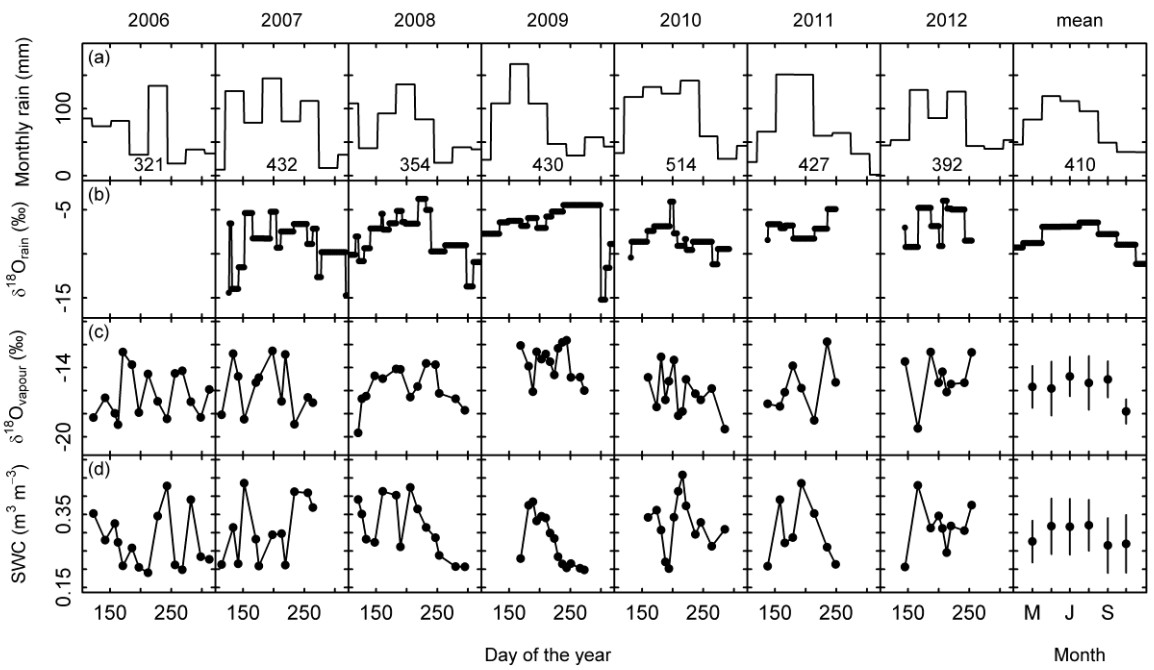

**Figure 1: Multi-seasonal (2006-2012) and average patterns of monthly rainfall sums (a), $\delta^{18}$O of rain ($\delta^{18}$O$_{rain}$) (b), $\delta^{18}$O of atmospheric vapour ($\delta^{18}$O$_{vapour}$) (c), and volumetric soil water content (SWC, m$^3$ water m$^{-3}$ soil) at 7 cm depth as predicted by the standard MuSICA simulation (d). Permanent wilting point: 0.19 SWC; field capacity: 0.49 SWC. $\delta^{18}$O$_{rain}$ and $\delta^{18}$O$_{vapour}$ refer to**
10 **measurements at the experimental site during the vegetation and soil sampling. $\delta^{18}$O$_{rain}$ was determined following individual rains during the vegetation periods of 2007 to 2012. Rainfall data were taken from the DWD weather station of Munich airport, located at the same altitude ~3 km south of the experimental site. The rainfall amount in the main growing period of each year (May to August) is given at the bottom of each panel in (a). Groundwater, at ~1.5 m below the soil surface, had an average $\delta^{18}$O of 10.0‰ (±0.4‰ SD).**





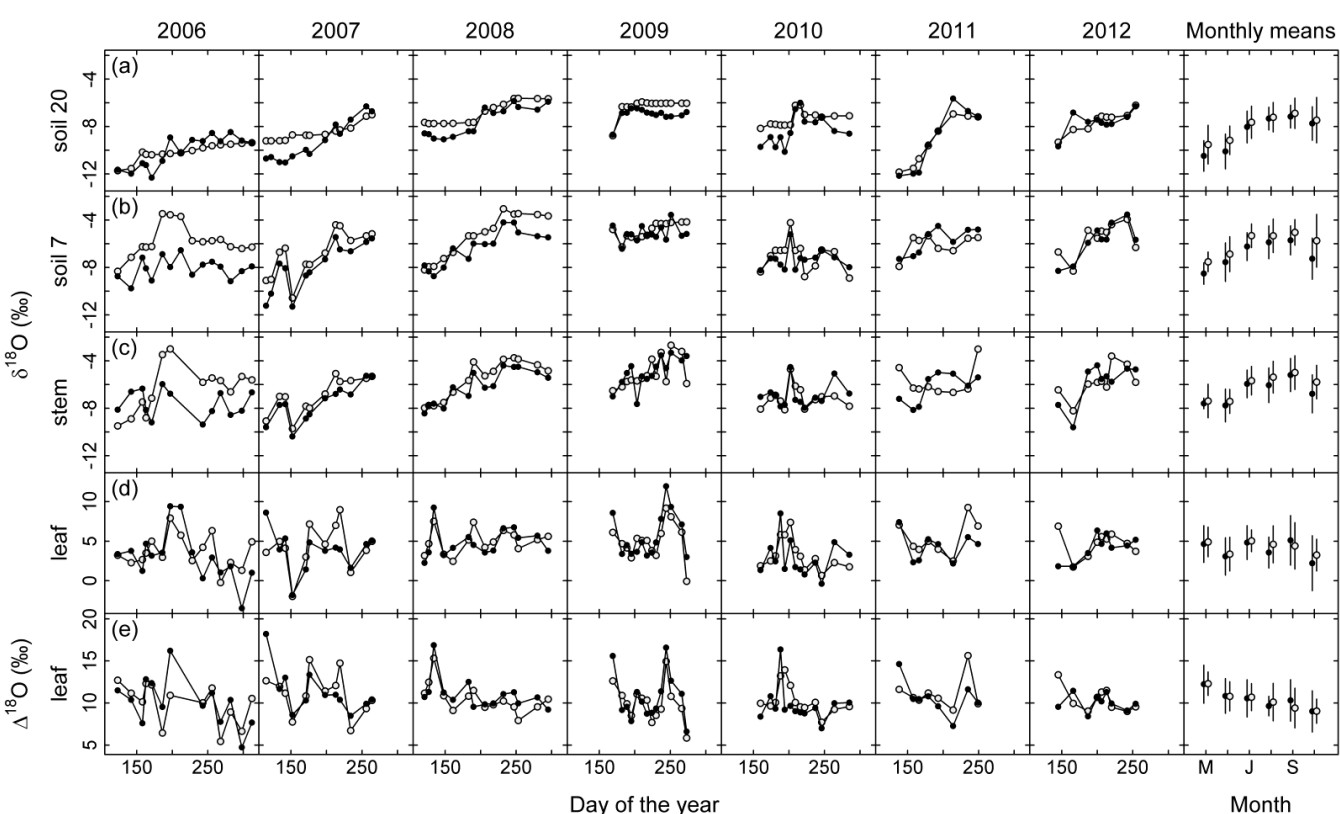

Fig. 2: Multi-seasonal (2006-2012) and monthly average variation of $\delta^{18}O$ in grassland ecosystem water pools: soil water at 20 (a) and 7 cm depth (b), stem (c) and leaf water (d), and $^{18}O$ enrichment ($\Delta^{18}O$) of leaf water (e), as observed (closed symbols) or predicted by the standard MuSICA simulations including a two-pool leaf water model (light gray). The parameters for the standard MuSICA simulations are given in the Supplement, Table S1). The error bar in the monthly mean data displays the standard deviation.





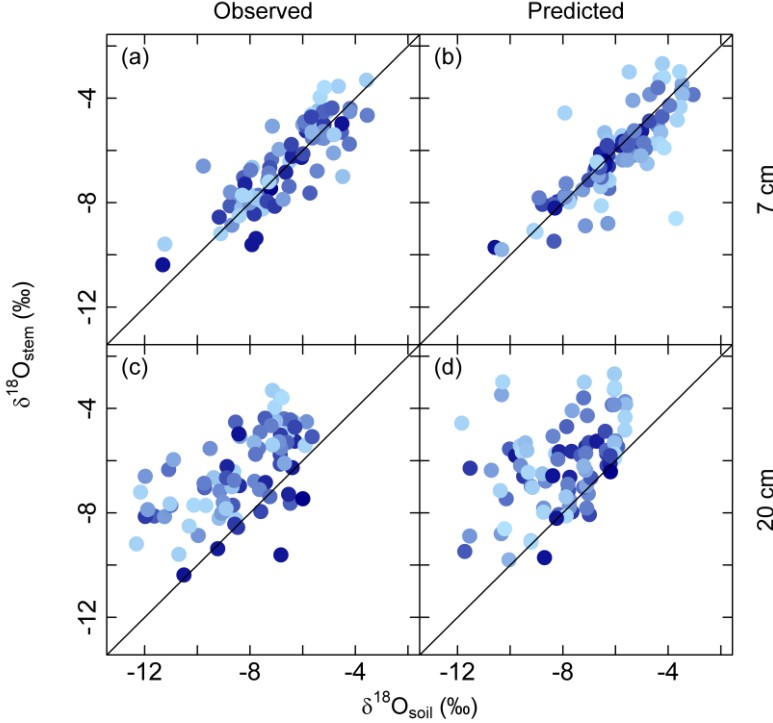

**Fig. 3: Correspondence between the $\delta^{18}O$ of stem water and soil water at 7 cm (observed, (a) and predicted, (b)) and at 20 cm depth (observed, (c) and predicted, (d)). Colour strength indicates soil water content at 7 cm depth as predicted by MuSICA with standard parameterisation: light blue, dry soils; dark blue, soils near field capacity (for colour coding to SWC scale, see Fig. 4). The $R^2$ for the relationship between $\delta^{18}O_{stem}$ and the $\delta^{18}O_{soil}$ at 7 and 20 cm depth was $R^2$ = 0.69 and 0.34 for the observed data comparison and $R^2$ = 0.65 and 0.17 for the modelled-modelled relationship. The straight lines represent the 1:1 relationship.**



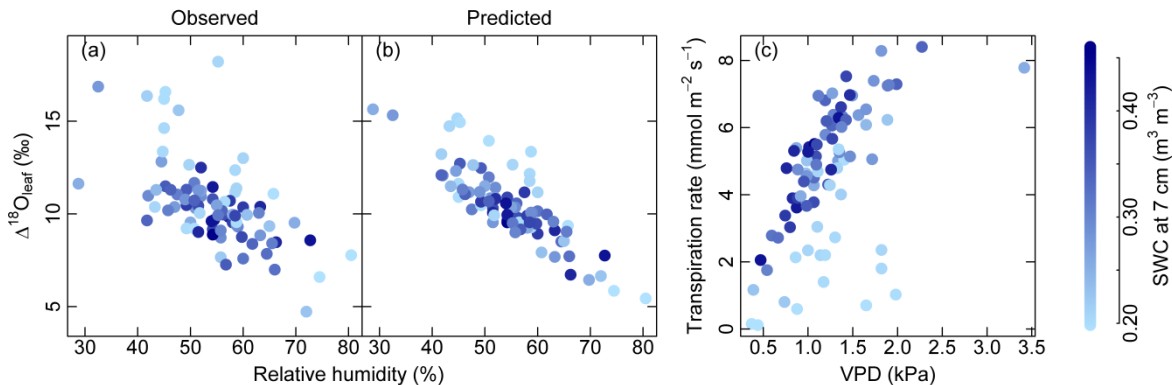

**Fig. 4: Relationship between relative humidity of air (RH) and observed Δ¹⁸O_leaf (a) and predicted Δ¹⁸O_leaf (b), and modelled response of transpiration to observed vapour pressure deficit (VPD) (c). Strength of blue colour from light to dark indicates the soil water content (SWC) at 7 cm depth as predicted by MuSICA with standard parameterisation. Permanent wilting point: 0.19 SWC; field capacity: 0.49 SWC. Predicted Δ¹⁸O_leaf data and transpiration rates were obtained with MuSICA in standard parameterisation and a two-pool leaf water model. Multiple regression analysis revealed effects of both RH and SWC on Δ¹⁸O_leaf (see Table 4).**

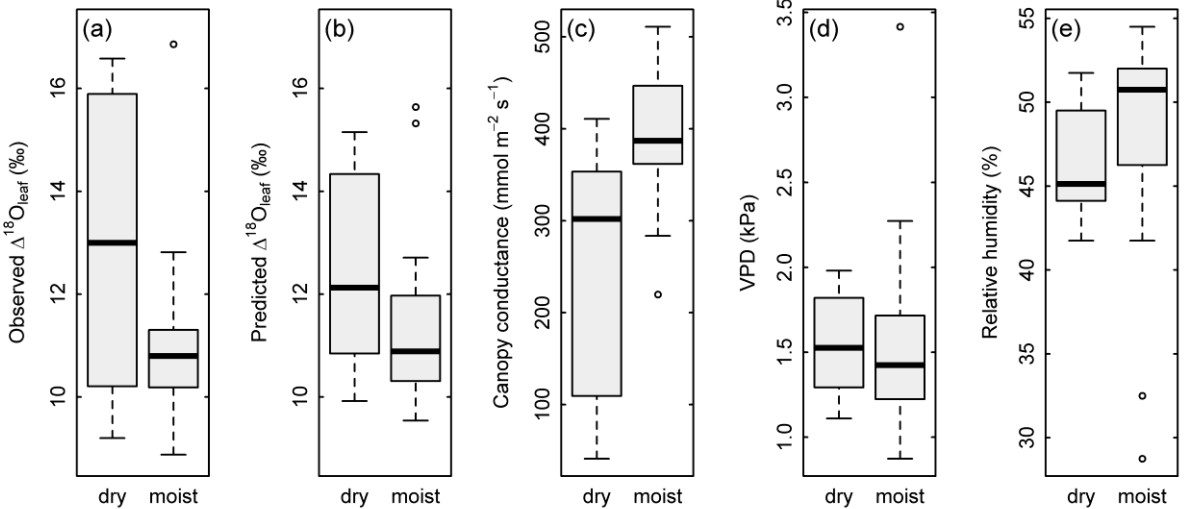

**Fig. 5: Boxplots showing the effect of soil water content ('dry' in comparison with 'moist') on observed Δ¹⁸O_leaf (a), predicted Δ¹⁸O_leaf (b), and modelled canopy conductance, $g_{canopy}$ (c) under conditions of low air relative humidity (<55% RH). Differences between dry and moist soil conditions were significant at P=0.03 (a), 0.06 (b) and 0.003 (c). At the same time, observed air VPD (d) and relative humidity (e) did not differ between dry and moist soil for the displayed subset (RH < 55%). Dry soil was defined as <0.25 SWC ($n = 12$), moist soil as ≥0.25 SWC ($n = 29$) at 7 cm depth. With a permanent wilting point of 0.19 SWC and a field capacity of 0.49, a SWC <0.25 corresponded to less than 25% of the maximum plant-available water at 7 cm.**



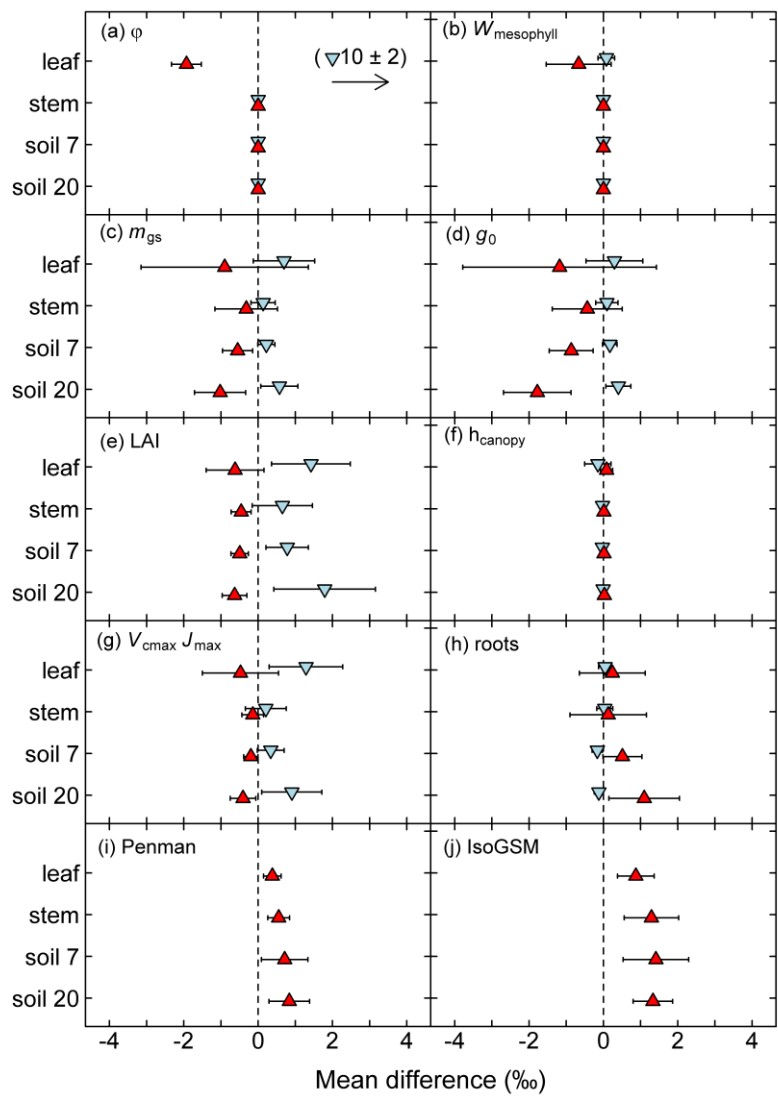

**Fig. 6: Sensitivity of modelled midday δ¹⁸O of leaf, stem and soil water at 7 and 20 cm depth to various parameters of the MuSICA model. The sensitivity was tested by varying one parameter while keeping all other parameters the same as in the standard MuSICA parameter set (Table S1), and was quantified as mean difference from the reference run and the standard deviation of the differences, displayed by error bars (see text). Parameter identity is given in the upper left corner of each panel. In (a) to (h), blue down-pointing triangles refer to the low parameter value, red up-pointing triangles to the high parameter value of a sensitivity run, as given in the Materials and Methods. In (i) the Moldrup submodel for the water vapour effective diffusivity in the soil was replaced by the Penman model. In (j) we used IsoGSM-predicted δ¹⁸O$_{rain}$ and δ¹⁸O$_{vapour}$ data instead of locally determined δ¹⁸O$_{rain}$ and δ¹⁸O$_{vapour}$ data for the isoforcing of MuSICA. Note that the low parameter value for Péclet number (a) predicted a far greater deviation of δ¹⁸O$_{leaf}$ than any other parameter.**





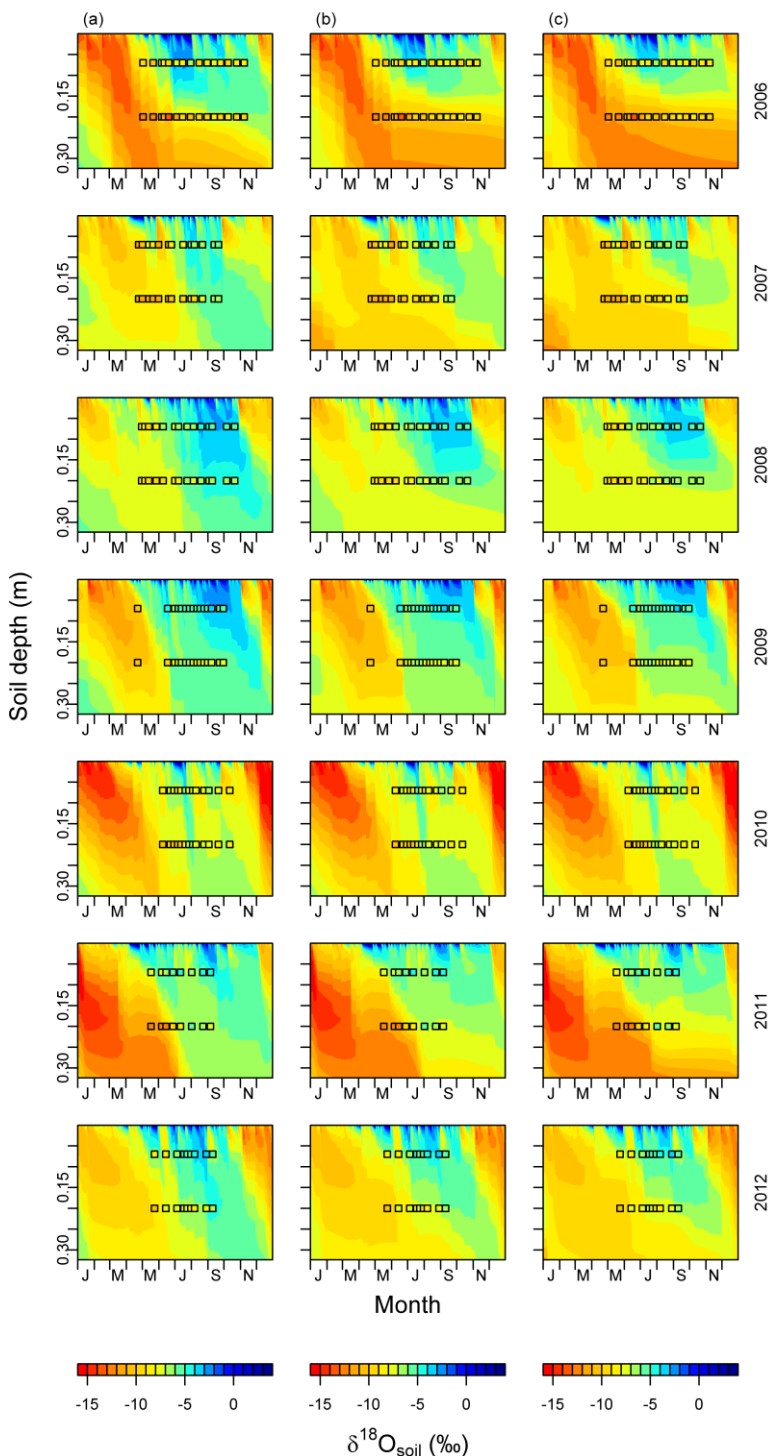





**Fig. 7: Soil water $\delta^{18}$O dynamics predicted for the studied period (2006-2012) with (a) low, (b) intermediate, and (c) high $V_{cmax}$ and $J_{max}$. Values for low and high parameter values are given in the Materials and Methods. Observed values for $\delta^{18}$O$_{soil}$ at 7 and 20 cm are displayed by squares. The same colour scheme is used for predicted and observed values and for each year and scenario. The abbreviations on the x-axes indicate the months.**



**Fig. 8: Flowchart illustrating how changes in photosynthetic parameters ($V_{cmax}$ and $J_{max}$) affect soil water content (SWC) and isotopic composition ($\delta^{18}$O$_{soil}$).**