# Peer review of "The 18O ecohydrology of a grassland ecosystem – predictions and observations"

_Hydrology and Earth System Sciences, 2019_

## Referee Comment (RC1) · Matthias Beyer (Referee) · 6 Mar 2019

Thank you for letting me review the manuscript 'The 18O ecohydrology of a grassland ecosystem –predictions and observations'. I enjoyed reading. In their work, the authors apply an 18O-enabled soil-plant-atmosphere transfer model in order to predict the dynamics of $\delta$18O in soil water, the depth of water uptake, and the effects of soil and atmospheric moisture on 18O-enrichment of leaf water in a grassland in southern Germany. In particular, they investigate the propagation of the $\delta$18O signal of rainwater through soil water pools, root water uptake and 18O enrichment of leaf water by tracing, predicting and validating $\delta$18Osoil, $\delta$18Ostem and $\Delta$18Oleaf. Finally, the authors test two models for describing $\Delta$18Oleaf at the canopy scale (the two-pool model or the Péclet model) and evaluate their performance.

Without doubt, this manuscript is well-prepared and written. The structure is clear, research questions are stated concisely, and the introduction provides a thorough overview on the topic. The graphics are suitable and well illustrated. I also agree to the authors that the model results are promising. The applied model MuSICA definitely seems capable of simulating ecohydrological processes including water isotopes. In my opinion, the hydrological and ecological community definitely needs a more integrated approach in modeling and investigating, and MuSICA seems a promising approach to that. I do not have major criticism on the manuscript, but a number of questions and comments that should be addressed in a revised version.

In summary those are: - In general, I find that the discussion of the results needs to be more critical. Yes, the results are good for an uncalibrated model. BUT: Grass is (sorry for saying that) probably the simplest plant to model (homogeneous and short roots). Looking at the isotope results, the 20cm depth and also under dry circumstances does not really fit well – see $R^2$. Hence, I would appreciate a more critical discussion, you have to highlight also the weaknesses that certainly still exist. Also, a total water balance is always a good means of validation and would be nice to have.

- The results section contains a lot of discussion (see detailed comments) - Why was model not calibrated?

- Why was 2H not used? How was fractionation evaluated without 2H - did the authors simply use the offset of 18O from the LMWL? Is the model capable of modeling 2H as well? The dual-isotope space enables a more comprehensive understanding of processes. Also, it is more sensitive compared to 18O and since the authors did a sensitivity study, perhaps very useful. I don't say I expect that in a revised version, but I am interested on the authors opinion on that.

Having that said, I suggest minor revision. I am looking forward to see the manuscript published in HESS.

Detailed comments:

[Figure]

Abstract l.20: grazing pressure, but how about rooting depth? Grasses are shallow-rooted so any other uptake is not expected?!

l.20: respond to atmospheric moisture....does that mean leaves take up moisture from the atmosphere? (foliar uptake???)

l.21: two non-mixing pools....is that realistic or justified?

l.26: the second sentence is not well written/unconcise

l.29: explain better or provide citation – explain why do leaves fractionate

p. 2 l.14: 'source water' for plants would be soil or groundwater, but not xylem water as it is plant water already

p. 2 l.15/16: 'summer' and 'winter' should be related to the particular study area, these statements are not true for the whole earth....

p. 2 l.29: 'enrichment above ....' I know what you mean but this is written ambiguous – stem water can also be subject to fractionation under certain conditions. It should be more clearly expressed what is meant with this sentence.

p. 2 l.31: 'many authors' – could you provide some citations, please?

p. 3 ll.2-14: this is well written!

p. 3 l.15: is this relevant for grasslands only?

p.4.l.5: please review this sentence and provide more information...which species, which soil depths, what exactly is meant with 'growing season'

p.5.l8: though you cite a paper on the cryogenic system you use, it would be nice to specify temperature and extraction time here

p.6.ll. 1 & 2-7: These information belong together, I'd suggest to either put the first part down or the second up

p.7 l. 33: based on what was the beta distribution assumed (based on previous research or citation) p.10.l 2: Why does the ratio need to remain 1.6?

p.10.ll. 4-6: Perhaps that fits better to 2.4.1 isoforcing

p.10.l21: Was predicted soil water content validated somehow?

p.11. l 29: in the way that (word missing)

p.11: paragraph 3.4 contains a lot of discussion, I suggest reviewing and removing some of the 'judging' (e.g. last sentence or l.29/30)

p.12.l.21: MLR does not appear in the methods/statistics

p.12.l.23: weakly significant? I think this should be rephrased → significant or not

p.12. paragraph 3.5.: the authors mix VPD and relative humidity quite a lot here, which makes this chapter hard to read. I suggest restructuring and rephrasing of this chapter (though the results completely make sense)

p.13l 4-10: Discussion

p.13. l.26-32: This sounds more like a conclusion

p.14. l.5: quite

p.14. l.6-7: suggest rephrasing: 'likely result from sampling effects and analytical error'

p.14. l.12-23: I agree, but also it should be clear that grass with a fairly uniform uptake depth right below surface is probably the easiest of plants to model. This is not a criticism but would be interesting how the model performs for different plant types.

4.2: I am not sure if this deserves an own chapter. I believe that it is true that the grass takes the water mainly from the upper depths but considering the characteristic shape of soil water isotope profiles at the surface (enrichment and subsequent decrease of isotope values towards a constant value), the used resolution of only 2 depths might not reveal true uptake patterns. Also see Rothfuss and Javaux, 2016.

p.15. l.26-27: 'online transpiration isotope method' this appears here for the first time?

p.16 l.9-11: I like this chapter, but the last sentence does not make sense – why compare and justify grass species with a study on non-grass-species?

Conclusions: An experienced and known Professor once gave me the advice 'A good paper doesn't need a conclusion – the reader draws it him/herself.' The authors should decide themselves, but I feel emphasizing some key points in the manuscript/abstract a bit more would be sufficient without conclusion.

Fig. 3: As stated above, the model does not work that well for 18O. I think this needs to be discussed thoroughly

Rothfuss, Y., Javaux, M., 2016. Isotopic approaches to quantifying root water uptake and redistribution: a review and comparison of methods. Biogeosciences Discuss. 1–47. https://doi.org/10.5194/bg-2016-410

---

## Referee Comment (RC2) · Anonymous Referee #2 · 15 Mar 2019

Hirl and coauthors present an impressive data set of seven years of isotopic observations in a grassland and an equally impressive modelling effort of the data.

The interpretation of the data is regrettably only discussing the isotopes and gives very little insight into the water fluxes of the ecosystem.

For example, if main water uptake is always at 7 cm depth even when this layer falls dry, then ecosystem transpires probably less than possible during this times because it would have access to more water in deeper soil. How is the ecosystem reacting? Is it shutting down the stomata? Is it changing its carboxylation capacity and stomata close thence? Or both? And why would a grassland do this? I guess it is well established in trees that they would harvest deeper soil water.

[Figure]

Are any of the other variables telling me something about the ecophysiology of the plants or the ecohydrology of the ecosystem? Are leaf water isotopes telling me something? They tell me at least that there is nighttime conductance. Is there also nighttime transpiration? Anything else?

I think, therefore, that the claim in the conclusions that the "work highlights the usefulness of mechanistic 18O-enabled modelling for explorations and quantitative analyses of the ecohydrology of ecosystems." is premature because only point (2) of the three points, i.e. root water uptake is actually ecohydrology of the ecosystem. The other points are about 18O ecohydrology, as the title of the paper suggests.

I have to admit that I had problems with the sensitivity analysis. Firstly, the mean difference is not a good measure. Differences can cancel out even when the model reacts strongly to a change. Most people use variance, standard deviation or root mean square error to avoid this. I guess that would be something like the error bars in Fig. 6. Secondly, one can of course use "arbitrary" ranges of model parameters to look at the output range, but then one cannot compare anymore the output ranges between the different parameters as done in Fig. 6. One wants to disturb each parameter similarly. So a derivative would probably be a good idea, or an elasticity.

Lastly, the authors suggest that there is no Péclet effect but rather a second unenriched water pool. While the data seem to support this, I would have expected a much better discussion.

I cannot find any mentioning of the 2D formulation of Farquhar and Gan (2003) while this should probably be the correct model. For example, what would be the effect if the leaf followed exactly this 2D model but the leaves were sampled only partly, not

sampling the least enriched part ?

The very small discussion starts with the possibility of xylem (or associated tissues) water and non-steady state but then only talks about the latter. I would have loved to see insights about grass blade anatomy, especially from this group who knows it that well.

I also do not follow the argument that there is no non-steady-state effect in the missing correlation with transpiration because the model includes non-steady state. The model yes, the data no. Margaret Barbour's group also claimed to see no Péclet effect but if they plotted their data against the isotopic composition of transpiration rather than xylem, the Péclet effect re-emerged.

The data sampled 7 species while the model describes one mean species. What is the effect of this? Could an averaging of different leaf dynamics not lead to the observed missing correlation with transpiration?

Overall I compliment the authors on this very nice data set and the very careful modelling, and wish to see the paper published soon.

---

## Author Comment (AC1) · 18 Apr 2019

**Reply to reviewer 2**

In what follows, we respond to the individual comments and recommendations of reviewer 2, R2. These responses are keyed to the specific comment by numbering, and are given in blue print, followed by indications of the changes made in the manuscript (in italics), and referring to the position in the original manuscript. Also, we revised again the entire manuscript for clarity, paying close attention to all of the reviewers' comments.

**R2 1**
Hirl and coauthors present an impressive data set of seven years of isotopic observations in a grassland and an equally impressive modelling effort of the data.

We thank the reviewer for the encouraging and thought-provoking comments that helped us much to improve the presentation of our work.

**R2 2**
The interpretation of the data is regrettably only discussing the isotopes and gives very little insight into the water fluxes of the ecosystem.

Yes, our results and interpretations centre on $^{18}O$ of water in the different ecosystem components, although we do present model predictions of canopy conductance and transpiration as a function of soil water content and leaf-to-air vapour pressure deficit (Fig. 4c and 5c), we compare measurements and predictions of latent heat flux (Fig. S1), and we make predictions of the soil depth distribution of water contents and root water uptake (Fig. S12). This approach was dictated by the main focus of the work that consisted in systematically evaluating our (eco)system-scale understanding of the propagation of the $\delta^{18}O$ signal of rainwater through soil water, root water uptake and $^{18}O$ enrichment of leaf water (as specified in the Introduction, P3 L30ff), by comparison of model predictions and observations. In that sense, our work is 'restricted' to the $^{18}O$-ecohydrology of the system, as we explore how the different hydrological properties of the system (given by the parametrization of the model) dictate the dynamics of $\delta^{18}O$ of water with depth in the soil, water taken up by the vegetation, and enrichment in the leaves. We believe that this is the most novel aspect of this work, and also the topic that we can support/validate/evaluate best with observations. Thus, our paper demonstrates how knowledge of $\delta^{18}O$ of distinct water pools can help us to assess the ability of a locally-parameterized $^{18}O$-enabled mechanistic soil-plant-atmosphere model in predicting the hydrology of a system. For instance, the fact that the $^{18}O$-enabled hydrology inside MuSICA predicted well the observed $^{18}O$-dynamics at different depths in the soil and in the water taken up by the root system indicates strongly that the ensemble of model parameters also predicted well the spatio-temporal dynamics of soil water contents (including emptying and refilling dynamics) and depth distribution of root water uptake. We hope that this approach – when developed further – can also be helpful later on for the ecohydrological interpretation of $\delta^{18}O$ in biological archives (e.g. $\delta^{18}O_{cellulose}$ extracted from Park Grass Experiment hay samples originating from the last century).

To address the point in the revision, we
-        added the definition of the term $^{18}O$-ecohydrology and its' objectives and potential (P2 L5). *"This science, that explores relationships between the spatio-temporal dynamics of water in the soil-vegetation-atmosphere system with help of the dynamics of $\delta^{18}O$ of water in its different components, may be termed $^{18}O$ ecohydrology."*
-        improved and expanded the discussion/interpretation of soil water dynamics and root water uptake (see below, and responses to reviewer 1, Matthias Beyer).

**R2 3**
For example, if main water uptake is always at 7 cm depth even when this layer falls dry, then ecosystem transpires probably less than possible during this times because it would have access to more water in deeper soil. How is the ecosystem reacting? Is it shutting down the stomata? Is it

changing its carboxylation capacity and stomata close thence? Or both? And why would a grassland do this? I guess it is well established in trees that they would harvest deeper soil water.

Yes, these are important points, that we address in 4.2 (revised, see below). See also responses to MB and relevant changes made, above.

Being restricted to only 2 depths, the spatial resolution of our observations of $\delta^{18}O_{soil}$ is limited, and there are methodical issues on the precision for estimation of the depth of root water uptake from such observations alone. Here, the (locally parameterized) hydrological model inside MuSICA does help. This predicted that root water uptake occurred over a broader zone (Fig. S12), with a mean (uptake-weighted) depth of root water uptake above a soil depth of 15 cm in 90% of all sampling dates (new Fig. S13).

We had no observations of stomatal conductance and carboxylation capacity, that would allow us to address their responses to drying soil. However, the model did consider an effect of soil drying on stomatal conductance (dependent on predawn leaf water potential) (P7 L22-24). The predicted effect of that is displayed in Fig. 5c. The sensitivity analysis did show that predictions of $\delta^{18}O_{soil}$ at the different depths was responsive to stomatal conductance. Therefore, the generally good agreement between observed and predicted $\delta^{18}O_{soil}$ did suggest that the ensemble of (photosynthetic and hydrological) model parameterization predicted the spatio-temporal variation in SWC and root water uptake quite well.

Interestingly, the model also predicted that SWC were occasionally lower below 25 cm than above that depth, particularly when rainfall recharged the top soil, but was insufficient to recharge the soil at greater depths (Fig. S12). Such phenomena occurred relatively frequently in the second half of the growing season. That fact could contribute additionally to explain why root water uptake occurred mainly from shallow soil depths (i.e. <20 cm below soil surface).

Certainly, the shallow root distribution also dictated a shallow depth of root water uptake. That shallow root distribution probably resulted from morpho-physiological constraints, particularly in the grasses and white clover (which comprised about 90% of the total pasture vegetation): in these species, adventitious roots compose virtually the entire root system, and root turnover is rapid and connected with leaf turnover at phytomere level (Yang et al., 1998; Robin et al., 2010) and assimilate supply to roots is reduced when grazing pressure is high (e.g. Bazot et al., 2005). In addition, the extremely high nutrient demand of frequently-defoliated vegetation is another factor that contributes to explain the formation and maintenance of a very shallow root system, as virtually all nutrient returns (mainly excreta from the grazing cattle) occur superficially.

We revised the entire manuscript for clarity concerning the above issues and, particularly, revised rigorously the first part of section 4.2, which now reads:

*"The comparison of observed $\delta^{18}O_{stem}$ and $\delta^{18}O_{soil}$ (Fig. 3a) strongly suggested that root water uptake occurred mainly at shallow depths (<20 cm) throughout the vegetation periods, largely independently of changes in SWC. That interpretation of observed data was based on comparison of $\delta^{18}O_{stem}$ and $\delta^{18}O_{soil}$ at two depths (7 and 20 cm) only, which provides limited spatial resolution and cannot inform precisely on the depth of root water, if $\delta^{18}O_{soil}$ does not change monotonously with soil depth (Rothfuss and Javaux, 2017; Brinkmann et al., 2018). Such information can be improved by a locally-parameterized, physically-based, $^{18}O$-enabled ecohydrological model, as shown here. For instance, the standard MuSICA runs (Fig. 3b) indicated near-monotonous increases of $\delta^{18}O_{soil}$ between 20 and 7 cm depth, matching well the observations in the majority of sampling dates (Fig. S13). Further, the simulations predicted a mean (uptake-weighted) depth of root water uptake at <15 cm in 90% of all sampling dates, independently of SWC and observations of $\delta^{18}O_{soil}$. Support came also from the MuSICA sensitivity analysis (Fig. 6h) in showing that $\delta^{18}O_{stem}$ was well predicted by the model only when root length density was maximum at shallow soil depth. The potential range of rooting depths is large in grassland, depending on site, species, climatic and management effects (Schenk and Jackson, 2002; Klapp, 1971). So, why was root water uptake constrained to shallow depths in this drought-prone permanent grassland system? Several factors likely contributed: (1) the shallow top-soil overlying calcareous gravel (Schnyder et al., 2006), (2) the rapid shoot and root biomass turnover, that is associated with high phytomer dynamics leading to short leaf and root lifespan in intensively managed grassland (Schleip et al., 2013; Yang et al., 1998; Auerswald and Schnyder, 2009; Robin et al., 2010), (3) the high rates of shoot tissue (mainly leaves) losses that elicit a priority for assimilate*

*(including reserve) allocation to shoot regeneration at the expense of the root system (e.g. Bazot et al., 2005), and (4) predominant placement of the root system near the soil surface dictated by the high need for nutrient interception and uptake (e.g. from excreta deposits), to compensate the high rates of nutrient losses due to grazing (Lemaire et al., 2000). Importantly, (5) in a relatively high number of cases, the model predicted situations in which rainfall recharged mainly the top soil, while SWC at depths >20 cm remained low (e.g. June-end of year 2006, April-October 2007, or May-end of year 2008; Fig. S12; see also below). Principally, however, factors (2)-(4) alone can explain why shallow rooting depth is a typical feature of intensively grazed grasslands (Troughton, 1957; Klapp, 1971). Also, Prechsl et al. (2015) did not find an …”*

**R2 4**
Are any of the other variables telling me something about the ecophysiology of the plants or the ecohydrology of the ecosystem? Are leaf water isotopes telling me something? They tell me at least that there is nighttime conductance. Is there also nighttime transpiration? Anything else?

In the main, the ecophysiology of the plants and the ecohydrology of the ecosystem is reflected in the parameterization of vegetation and soil in MuSICA (Methods S2, Table S1, Figure S5, S6, S8), with many parameter values obtained from local measurement. The spatio-temporal dynamics of root water uptake (Fig. S12), and canopy conductance (Fig. 5c) and transpiration rate (Fig. 4c) at midday predicted by MuSICA are a result of that parameterization.
And yes, the diurnal $\delta^{18}O_{leaf}$ data indicate that stomates were not completely closed during the night (P7 L18-19), a factor that was reflected in the parameterization of MuSICA (Table S1). Yet, predicted night-time transpiration (estimated by latent energy flux) was always very low, in agreement with the eddy flux data (Fig. S1) and the generally high nocturnal relative humidity.
We did not have the detailed ecophysiological and ecohydrological observations to validate those specific predictions. However, we did validate MuSICA for the evapotranspiration (i.e. latent heat flux) predictions, and estimations of plant-available soil water in the entire top-soil (see also changes made in response to reviewer 1).
Most importantly, the good agreement between observed and predicted $\delta^{18}O$ in soil (at 7 and 20 cm depth), stem and leaf water does indicate that the model described the ecohydrology of the grassland system well.

In the revision, we added several sentences and phrases, clarifying those points (see also responses to reviewer 1):

P5 L27ff: *“The model was validated with latent energy flux (LE) data obtained from an eddy covariance station (EC) at the site. According to that comparison (Fig. S1), MuSICA estimates were unbiased ($LE_{MuSICA} = 0.997\ LE_{EC}$; $R^2 = 0.59$). Further, we compared MuSICA predictions of total plant-available soil water (PAW, mm) in the entire top soil with PAW modelling and data for the same site presented in Schnyder et al. (2006). For the 2007-2012 data, this yielded the relationship $PAW_{MuSICA} = 0.99\ PAW_{Schnyder\ et\ al.\ 2006} + 7.8$ ($R^2$ 0.83).”*

P7 L20ff: *“Although the diurnal pattern of $\delta^{18}O_{leaf}$ (Fig. S7) indicated some nocturnal stomatal conductance, the model generally predicted very low nighttime transpiration, in agreement with the eddy flux data (Fig. S1) and the generally high nocturnal relative humidity.”*

P 14 L12ff: *These ecohydrological processes are described explicitly in MuSICA, and agreement between observations and predictions of $\delta^{18}O_{stem}$ and $\delta^{18}O_{soil}$ at 7 and 20 cm depth indicates that MuSICA is capable of simulating these ecohydrological processes including $^{18}O$ of the different water pools.*

And P15 L4ff: *“Predictions of $\delta^{18}O_{soil}$, particularly below the main zone of most water uptake, at 20 cm, were influenced markedly by estimates of LAI…”*

**R2 5**

I think, therefore, that the claim in the conclusions that the "work highlights the usefulness of mechanistic 18O-enabled modelling for explorations and quantitative analyses of the ecohydrology of ecosystems." is premature because only point (2) of the three points, i.e. root water uptake is actually ecohydrology of the ecosystem. The other points are about 18O ecohydrology, as the title of the paper suggests.

We understand the criticism, which is – we believe – partly due to our omission of a clear definition of $^{18}$O-ecohydrology, and the objectives of its application in the present context.

In the revision, we added the definition in the Introduction. Here, we employed the ecohydrological model implemented in MuSICA to predict the $\delta^{18}$O of water at different soil depths, the $\delta^{18}$O of water taken up from the soil, and the $^{18}$O-enrichment in leaves. In that we also evaluated several methodical/conceptual, $^{18}$O-ecohydrological uncertainties impacting on such predictions, such as (1) the choice of the water vapour effective diffusivity in the soil (Moldrup vs Penman), (2) the source of the rain water and atmospheric vapour $\delta^{18}$O (local data vs IsoGSM estimations), and (3) alternative leaf water-$^{18}$O-enrichment models (two-pool vs Péclet). The capability of the model to predict the $\delta^{18}$O of the different water pools then indicates that the model is equally capable to predict the different ecohydrological processes (that underlie the $\delta^{18}$O predictions and observations).

Also, we revised all text carefully to eliminate any opportunity for misunderstandings. In particular, we highlight that a physically-based $^{18}$O-enabled ecohydrological model (as implemented in MuSICA) can provide insight in ecohydrological processes, such as the spatio-temporal dynamics of soil water and root water uptake, and transpiration or canopy/stomatal conductance. Concerning the latter, we find it interesting that dry soil conditions (under similar atmospheric conditions) led to increased $^{18}$O-enrichment (on average) in both the observed and predicted data, although evidence for a Péclet effect was missing in our data.

In the revision, we made the following main corrections, additions and deletions:
Abstract
P1 L16: "*Using the ecohydrology part of a physically-based, $^{18}$O-enabled soil-plant-atmosphere transfer model (MuSICA), we evaluated our ability to predict the dynamics ...*"
P1 L18: "*The model accurately predicted the $\delta^{18}$O dynamics of the different ecosystem water pools, suggesting that the model generated realistic predictions of the vertical distribution of soil water and root water uptake dynamics. Observations and model predictions indicated that water uptake occurred predominantly from shallow (<20 cm) soil depths ...*"

Introduction
P2 L5: "*This science, that explores relationships between the spatio-temporal dynamics of water in the soil-vegetation-atmosphere system with help of the temporal dynamics of $\delta^{18}$O of water in its different components, may be termed $^{18}$O ecohydrology*".

Conclusion
We deleted the Conclusions section (see also response to reviewer1, MB 39)

**R2 6**
I have to admit that I had problems with the sensitivity analysis. Firstly, the mean difference is not a good measure. Differences can cancel out even when the model reacts strongly to a change. Most people use variance, standard deviation or root mean square error to avoid this. I guess that would be something like the error bars in Fig. 6. Secondly, one can of course use "arbitrary" ranges of model parameters to look at the output range, but then one cannot compare anymore the output ranges between the different parameters as done in Fig. 6. One wants to disturb each parameter similarly. So a derivative would probably be a good idea, or an elasticity.

We understand the point raised by the reviewer. We realize that our description of the sensitivity analysis and of Fig. 6 lacked some precision, and we improved that in the revision (see below).
We like to emphasize that our sensitivity analysis presents two types of (sensitivity) information/variables: (1) the mean sensitivity to a change of a parameter value (upper or lower value) on the metric of interest (e.g. $\delta^{18}O_{leaf}$), in relation to the standard simulation, as shown on the x-axis as

'mean sensitivity', and (2) the standard deviation of the sensitivity (given by the error bar). The standard deviation captures the variability of the response to a parameter change between the individual sampling occasions. If cancelling effects result from the change of a parameter value, resulting in a mean sensitivity close to zero, that cancelling behavior is revealed by the (size of the) standard deviation of the sensitivity (e.g. the effect of the upper parameter value on $\delta^{18}O_{leaf}$ in panel 6h). Panel 6a reports a very different type of behavior, as changing the parameter value caused no cancelling effects on $\delta^{18}O_{leaf}$ (as was indicated by the small standard deviation of the sensitivity), but a strong change in the mean sensitivity for $\delta^{18}O_{leaf}$. So, there were instances where changes of parameter values caused a 'general' effect (causing a positive or negative mean sensitivity) and instances where there were strong cancelling effects (leading to a large standard deviation of the sensitivity). Both types of sensitivities can be gleaned from our presentation of parameter sensitivities.

Thus our sensitivity analysis revealed four different types of sensitivities: (a) strong mean sensitivities, with no or little cancelling (e.g. $\delta^{18}O_{leaf}$ in panel 6a), (b) mean sensitivities combined with strong cancelling effects (e.g. $\delta^{18}O_{leaf}$ in panel 6c), (c) no mean sensitivities resulting from strong positive and negative cancelling effects (e.g. $\delta^{18}O_{leaf}$ in response to the high parameter value in panel 6h), and (d) absence of a mean sensitivity without cancelling effects (e.g. $\delta^{18}O_{stem}$, $\delta^{18}O_{soil\ 7}$ and $\delta^{18}O_{lsoil\ 20}$ in panels 6a and 6b).

Although we like the idea of calculating elasticities, in principle, we did see some problems:
1)      The $\delta^{18}O$ values are not ratio-scaled (but interval-scaled) and the zero value (0‰) is not an absolute zero, resulting in problems when comparing parameter effects on the $\delta^{18}O$ of the different water pools.
2)      'Elasticity' quantifies the percentage change of the output variable in response to a given percent change in the input parameter. This does not consider if a given percent change in the input parameter is hydrologically or physiologically plausible or relevant (particularly when model sensitivity is compared for different parameters).
3)      It may not be possible to draw universally valid conclusions from the elasticity. In case of a non-linear response of the variable under study, elasticity depends on the extent of change of the parameter. Yet, varying parameters by the same percentages, e.g. by +50% and by -50%, in order to 'disturb each parameter similarly', would neglect morpho-physiological or system knowledge on the 'realistic' (or 'plausible') range of values for each parameter. So, changing a parameter by a certain percentage is likely a more arbitrary choice than the one that we have taken.
Point 3) is also valid for derivatives.

Regarding the second point of the reviewer "one can of course use "arbitrary" ranges of model parameters to look at the output range":
This is a point that we had discussed extensively, during the work and preparation of the submitted manuscript. In effect, we did not use arbitrary values. Instead, we chose the upper and lower parameter values based on the range of values observed at the site (LAI, canopy height, mesophyll water content), ranges dictated by physical constraints of the system (root distribution), the origin of the $\delta^{18}O_{rain}$ data (IsoGSM predictions as opposed to local measurements), or – where we did not have own measurements – based on the range found in the literature for grasses/grassland ($\varphi$, $m_{gs}$, $g_0$, $V_{cmax}$ and $J_{max}$). In that way we ascertained realistic and physiologically meaningful upper and lower parameter values in the sensitivity analysis. In a way, this also dictated that we refrain from calculating elasticities.

On the basis of these facts and considerations, we would like to retain the approach to sensitivity analyses presented in the original manuscript. However, we did take the reviewer's comment/concerns very seriously and improved the presentation and description of the approach. This included: renaming the 'mean difference' by '*mean sensitivity*' (which is more appropriate and illustrative) and standard deviation of the difference by '*standard deviation of the sensitivity*', and explaining the rationale for the choice of this specific form of sensitivity analysis.

The legend to Fig. 6 now reads:
*"Fig. 6: Sensitivity of modelled midday $\delta^{18}O$ of leaf, stem and soil water at 7 and 20 cm depth to various parameters of the MuSICA model. The sensitivity was tested by varying one parameter while*

*keeping all other parameters the same as in the standard MuSICA parameter set (Table S1), as detailed in 2.5. Sensitivity (parameter effect) was quantified by two variables: the mean (or average) sensitivity (in ‰) resulting from the change of a parameter value relative to the reference run, and the standard deviation of the sensitivity which captures the variability of the response to a parameter-change for the different sampling times (displayed by error bars.) Strong averaging (cancelling) effects resulting from the change of a parameter value are revealed by large standard deviations of sensitivities. Note that the sensitivity analysis revealed four different combinations of parameter effects: (a) strong mean sensitivities, without cancelling effects, (b) strong mean sensitivities superposed with strong cancelling effects, (c) small mean sensitivities resulting from strong cancelling effects, or (d) absence of sensitivities unrelated to cancelling effects. Parameter identity is given in the upper left corner of each panel. In (a) to (h), blue down-pointing triangles refer to the low parameter value, red up-pointing triangles to the high parameter value of a sensitivity run, based on the range of values observed at the site or – where such values were missing – the range of reported values for grasses or grassland in literature (see Materials and Methods). In (i) the Moldrup submodel for the water vapour effective diffusivity in the soil was replaced by the Penman model. In (j) we used IsoGSM-predicted $\delta^{18}O_{rain}$ and $\delta^{18}O_{vapour}$ data instead of locally determined $\delta^{18}O_{rain}$ and $\delta^{18}O_{vapour}$ data for the isoforcing of MuSICA. Note that the low parameter value for Péclet number (a) predicted a far greater deviation of $\delta^{18}O_{leaf}$ than any other parameter.*

The relevant section of 2.5 was revised accordingly (P9 L21ff):
*"Parameter effects (sensitivities) were quantified by two variables: (i) the mean sensitivity relative to the reference run, obtained as the mean differences from the reference run as $(\sum_{i=1}^{n} (\delta_{sens,i} - \delta_{ref,i}))/n$, with $\delta_{sens,i}$ the $\delta^{18}O$ of a given water compartment (leaf, stem, or soil at 7 or 20 cm depth) in a sensitivity run and $\delta_{ref,i}$ that in the reference run, for a day i; and (ii) the standard deviations of the sensitivity, obtained from the differences between $\delta_{sens,i}$ and $\delta_{ref,i}$. The latter illustrated how strongly the effect of a parameter varied between sampling days, and hence how strongly it depended on the conditions encountered on one specific day. Thus, the sensitivity variables (mean and standard deviation of sensitivity) reported if changes in parameter values caused systematic/general effects (shown by the mean sensitivity), or cancelling effects (shown by the standard deviations of the sensitivity ), or combinations, or lack of the two."*

Also, paragraph 3.6 and 4.2 were revised for consistency.

**R2 7**
Lastly, the authors suggest that there is no Péclet effect but rather a second unenriched water pool. While the data seem to support this, I would have expected a much better discussion.
I cannot find any mentioning of the 2D formulation of Farquhar and Gan (2003) while this should probably be the correct model. For example, what would be the effect if the leaf followed exactly this 2D model but the leaves were sampled only partly, not sampling the least enriched part?

We sampled the entire leaf blades and the entire exposed part of the growing leaf blade of grasses, (which was a minor component of the total sample), and trifoliate leaves of white clover. In the case of *Taraxacum officinale*, we included half a leaf blade, severed along the length of the midrib. With that sampling protocol we integrated (but did not resolve) the entire gradients of evaporation-related [18]O-enrichment that occurred within the individual leaf blades, permitting (and restricting us to) the use of the whole-leaf version of the [18]O-enrichment model used to evaluate the occurrence of a Péclet effect. With that protocol, it was not possible to use the theory presented in the 2D formulation of Farquhar and Gan (2003); hence we used the non-steady-state version of the Péclet model, which is equivalent to that used by Gan, Wong, Yang and Farquhar (2003) for their experimental whole-leaf data.

In the revision we improved the respective paragraph, which now reads (P4 L27ff): *"Each leaf sample included all leaf blades, including the exposed part of the growing leaf, but excluding senescing leaves (cf Fig. 1 of Liu et al., 2017) from each of two vegetative tillers of D. glomerata and 16 vegetative tillers of L. perenne, P. pratensis and P. pratense, one half of a leaf blade of T. officinale (with the latter severed along, but not including, the mid-vein) and two trifoliate leaves of T. repens. This*

*protocol ensured collection of the entire within-leaf evaporative $^{18}$O-gradient of all sampled leaf blade tissue of the different species.*"

**R2 8**

The very small discussion starts with the possibility of xylem (or associated tissues) water and non-steady state but then only talks about the latter. I would have loved to see insights about grass blade anatomy, especially from this group who knows it that well.

We did not collect data on the anatomy of sampled leaves, as this was impractical (see also response to R2 9, below).

**R2 9**

I also do not follow the argument that there is no non-steady-state effect in the missing correlation with transpiration because the model includes non-steady state. The model yes, the data no. Margaret Barbour's group also claimed to see no Péclet effect but if they plotted their data against the isotopic composition of transpiration rather than xylem, the Péclet effect re-emerged.

Yes, correct, the model included non-steady-state. A significant fraction of the observations originated from non-steady-state conditions, others appeared to be close to steady-state (Figure S9).

In the revision, we looked at the subset of observations that exhibited seemingly near-steady-state $^{18}$O-enrichment (about half the data) to verify additionally if the relationship between the proportional difference between observed leaf water $^{18}$O-enrichment ($\Delta^{18}O_{leaf}$) and evaporative site enrichment ($\Delta^{18}O_e$) predicted by the Craig-Gordon model ($\Delta^{18}O_{e,ss}$) would indicate the existence of a Péclet effect for that subset. Again, we did not observe evidence of such an effect.

In the revision, we deleted the sentence P12 L15-18, replacing it by: *"Also, the relationship between modelled transpiration rate and the proportional difference between the observed $\Delta^{18}O_{leaf}$ and $\Delta^{18}O$ predicted by the Craig-Gordon model (Fig. S11) was non-significant, revealing no evidence of a Péclet effect. This was also true, when investigating that relationship with a subset of the data that included only the leaves that exhibited near-steady-state $^{18}$O-enrichment. This subset was estimated using model output to identify the times when near-steady-state conditions were most likely, and included about half of the data (results not shown)."*

**R2 10**

The data sampled 7 species while the model describes one mean species. What is the effect of this? Could an averaging of different leaf dynamics not lead to the observed missing correlation with transpiration?

Yes, we also wondered if inability to detect a Péclet effect in the mixed-species leaf sample could have resulted from different leaf water and, hence, $^{18}$O-enrichment dynamics in the different species. As we could not answer that question with the data from our grassland ecosystem study, we included ancillary data obtained separately with *Lolium perenne* and *Dactylis glomerata* in different experiments in controlled conditions by Margaret Barbour. These species formed part of the mixed-species sample in our grassland ecosystem. The *L. perenne* data were based on destructive measurements of leaf water $^{18}$O-enrichment; conversely, the experiment with *D. glomerata* employed an online gas exchange and equilibrated leaf water method. In both cases, a Péclet effect was not apparent.

In the revision we expanded and improved the discussion of the putative causes for the absence of a Péclet effect or for our inability of detecting one (P15 L23ff):
*"...environmental conditions. We do not know if putative between-species differences in leaf water dynamics and associated $^{18}$O-enrichement, or any other morpho-physiological effects e.g. associated with leaf aging, could have led to a missing correlation between the proportional difference between measured leaf water $^{18}$O-enrichment and that predicted by the Craig-Gordon model (1 - $\Delta^{18}O_{leaf}/\Delta^{18}O_e$) and transpiration rate. For these reasons, we explored this question with separate studies of L.*

*perenne and D. glomerata, two species that also formed part of the present grazed grassland ecosystem. Again, these studies found no evidence for a Péclet effect, and supported the two-pool model, as there was no relationship between the proportional difference between measured leaf water enrichment and that predicted by the Craig-Gordon model ...”*

**R2 11**
Overall I compliment the authors on this very nice data set and the very careful modelling, and wish to see the paper published soon.

Thank you!

**References not included in the Discussion paper**

Auerswald, K. and Schnyder, H.: Böden als Grünlandstandorte, in: Handbuch der Bodenkunde, edited by: Blume, H.-P., Felix-Henningsen, P., Frede, H.-G., Guggenberger, G., Horn, R., and Stahr, K., Wiley-VCH, 31, Erg.Lfg., 1-15, https://doi:10.1002/9783527678495.hbbk2009003, 2009.

Bazot, S., Mikola, J., Nguyen, C., and Robin, C.: Defoliation-induced changes in carbon allocation and root soluble carbon concentration in field-grown *Lolium perenne* plants: do they affect carbon availability, microbes and animal trophic groups in soil?, Funct. Ecol., 19, 886-896, https://doi.org/10.1111/j.1365-2435.2005.01037.x, 2005.

Brinkmann, N., Seeger, S., Weiler, M., Buchmann, N., Eugster, W., and Kahmen, A.: Employing stable isotopes to determine the residence times of soil water and the temporal origin of water taken up by *Fagus sylvatica* and *Pices abies* in a temperate forest, New Phytol., 219, 1300-1313, https://doi.10.1111/nph.15255, 2018.

Lemaire, G., Hodgson, J., de Moraes, A., and Nabinger, C.: Grassland Ecophysiology and Grazing Ecology, CABI Publishing, Wallingford, U.K., 2000.

Robin, A. H. K., Matthew, C., and Crush, J. R.: Time course of root initiation and development in perennial ryegrass – a new perspective, Pr. N. Z. Grassl. Assoc., 72, 233-240, 2010.

Rothfuss, Y. and Javaux, M.: Review and syntheses: Isotopic approaches to quantify root water uptake: a review and comparison of methods, Biogeosciences, 14, 2199-2224, https://doi:10.5194/bg-14-2199-2017, 2017.

Schenk, H. J. and Jackson, R.B.: Rooting depths, lateral root spreads and below-ground/above-ground allometries of plants in water-limited ecosystems, J. Ecol., 90, 480-494, https://doi.org/10.1046/j.1365-2745.2002.00682.x, 2002.

Yang, J. Z., Matthew, C., and Rowland, R. E.: Tiller axis observations for perennial ryegrass (*Lolium perenne*) and tall fescue (*Festuca arundinacea*): number of active phytomers, probability of tiller appearance, and frequency of root appearance per phytomere for three cutting heights, New Zeal. J. Agr. Res., 41, 11-17, https://doi:10.1080/00288233.1998.9513283, 1998.

---

## Author Comment (AC2) · 18 Apr 2019

**Reply to reviewer 1 Matthias Beyer**

In what follows, we respond to the individual comments and recommendations of reviewer 1, Matthias Beyer, MB. These responses are keyed to the specific comment by numbering, and are given in blue print, followed by indications of the changes made in the manuscript (in italics), and referring to the position in the original manuscript. Also, we revised again the entire manuscript for clarity, paying close attention to all of the reviewers' comments.

**MB 1**

Thank you for letting me review the manuscript 'The 18O ecohydrology of a grassland ecosystem – predictions and observations'. I enjoyed reading. In their work, the authors apply an 18O-enabled soil-plant-atmosphere transfer model in order to predict the dynamics of $\delta^{18}O$ in soil water, the depth of water uptake, and the effects of soil and atmospheric moisture on 18O-enrichment of leaf water in a grassland in southern Germany. In particular, they investigate the propagation of the $\delta^{18}O$ signal of rainwater through soil water pools, root water uptake and 18O enrichment of leaf water by tracing, predicting and validating $\delta^{18}O_{soil}$, $\delta^{18}O_{stem}$ and $\Delta^{18}O_{leaf}$. Finally, the authors test two models for describing $\Delta^{18}O_{leaf}$ at the canopy scale (the two-pool model or the Péclet model) and evaluate their performance.

We thank Matthias Beyer for the thorough and encouraging review and the detailed comments and recommendations that helped us much to improve the presentation of our work.

**MB 2**

Without doubt, this manuscript is well-prepared and written. The structure is clear, research questions are stated concisely, and the introduction provides a thorough overview on the topic. The graphics are suitable and well illustrated. I also agree to the authors that the model results are promising. The applied model MuSICA definitely seems capable of simulating ecohydrological processes including water isotopes. In my opinion, the hydrological and ecological community definitely needs a more integrated approach in modeling and investigating, and MuSICA seems a promising approach to that. I do not have major criticism on the manuscript, but a number of questions and comments that should be addressed in a revised version.
    In summary those are: In general, I find that the discussion of the results needs to be more critical.

We revised the discussion thoroughly, considering all points raised by the reviewer (see responses to individual comments, below).

**MB 3**

Yes, the results are good for an uncalibrated model. BUT: Grass is (sorry for saying that) probably the simplest plant to model (homogeneous and short roots).

We are uncertain if modelling grass is inherently much simpler than modelling a non-grass species. For instance, the potential range of rooting depths of perennial grasses (and other grassland plants) can be very large (up to 6 m depth; cf. Schenk and Jackson, 2002), and grazing pressure (or defoliation frequency) can affect rooting depth very strongly (e.g. Klapp, 1971, Figure 43, page 81), providing scope for a large variability in rooting depth and depth of water uptake in different grassland systems.

In the revision we added a paragraph in the discussion pointing to this factor (see MB 9, below).

**MB 4**

Looking at the isotope results, the 20cm depth and also under dry circumstances does not really fit well – see R2. Hence, I would appreciate a more critical discussion, you have to highlight also the weaknesses that certainly still exist.

We believe that there is some misunderstanding here, and revised the text to eliminate any opportunity for such misunderstanding (again, see responses to individual comments, below).

In fact, the model performance for predicting $\delta^{18}O_{soil}$ at 20 cm depth was really good, as was indicated by the close relationship of modelled and observed data ($R^2 = 0.79$) and the very small bias (MBE = 0.5‰; Table 2). Also, the observations and the model agreed rather well with respect to the relationship between $\delta^{18}O_{stem}$ and $\delta^{18}O_{soil}$ (Figure 3): that relation was close in both the observed ($R^2 = 0.69$) and predicted data sets (0.65) and virtually unbiased at a depth of 7 cm, independently of soil water contents. Further, the predictions and observations agreed in that both indicated a poor relationship between $\delta^{18}O_{stem}$ and $\delta^{18}O_{soil}$ at 20 cm, both in terms of scatter ($R^2 = 0.34$ for the observed and 0.17 for the model predicted relationships) and bias. On average, $\delta^{18}O_{stem}$ was ca 2‰ higher than $\delta^{18}O_{soil\ 20}$, meaning that $\delta^{18}O_{soil\ 20}$ did not agree with $\delta^{18}O_{stem}$. Thus, both the observations and the modelling independently indicated that water uptake must have occurred mainly from shallow depths (<20 cm).

In the revision, we worked through the text and relevant Table captions and Figure legends very carefully to enhance clarity and eliminate any ambiguity on model performance (see also response to MB 7, below).

The following main changes were made:

Abstract (P1 L18ff): "The model accurately predicted the $\delta^{18}O$ dynamics of the different ecosystem water pools, *suggesting that the model generated realistic predictions of the vertical distribution of soil water and root water uptake dynamics. Observations and model predictions indicated that water uptake occurred predominantly from shallow (<20 cm) soil …*"

P11 L14ff: "*Conversely, the relationship between $\delta^{18}O_{stem}$ and $\delta^{18}O_{soil}$ at 20 cm was generally weak, exhibiting large scatter and a significant offset between $\delta^{18}O_{stem}$ and $\delta^{18}O_{soil}$ at 20 cm for most of the data (Fig. 3c).*"

P11 L22ff: "MuSICA simulations were based on this assumption and reproduced very similar relationships between $\delta^{18}O_{stem}$ and $\delta^{18}O_{soil}$ as those observed at both depths, with *similar $R^2$, MBE and MAE* (Figs. 2-3), *thus showing a close agreement between observed and predicted data.*

P14 L25ff: The comparison of observed $\delta^{18}O_{stem}$ and $\delta^{18}O_{soil}$ (Fig. 3a) strongly suggested that root water uptake occurred *mainly* at shallow depths *(<20 cm)* throughout the vegetation periods, largely independently of changes in SWC. *That interpretation of observed data was based on comparison of $\delta^{18}O_{stem}$ and $\delta^{18}O_{soil}$ at two depths (7 and 20 cm) only, which provides limited spatial resolution and cannot inform precisely on the depth of root water, if $\delta^{18}O_{soil}$ does not change monotonously with soil depth (Rothfuss and Javaux, 2017; Brinkmann et al., 2018). Such information can be improved by a locally-parameterized, physically-based, $^{18}O$-enabled ecohydrological model, as shown here. For instance, the standard MuSICA runs (Fig. 3b) indicated near-monotonous increases of $\delta^{18}O_{soil}$ between 20 and 7 cm depth, matching well the observations in the majority of sampling dates (Fig. S13). Further, the simulations predicted a mean (uptake-weighted) depth of root water uptake at <15 cm, in 90% of all sampling dates, independently of SWC and observations of $\delta^{18}O_{soil}$. Support came also from the MuSICA sensitivity analysis (Fig. 6h) in showing that $\delta^{18}O_{stem}$ was well predicted by the model only when root length density was maximum at shallow soil depth. The potential range of rooting depths is large in grassland, depending on site, species, climatic and management effects (Schenk and Jackson, 2002; Klapp, 1971). So, why was root water uptake constrained to shallow depths in this drought-prone permanent grassland system? Several factors likely contributed: (1) the shallow top-soil overlying calcareous gravel (Schnyder et al., 2006), (2) the rapid shoot and root biomass turnover, that is associated with high phytomer dynamics leading to short leaf and root lifespan in intensively managed grassland (Schleip et al., 2013; Yang et al., 1998; Auerswald and Schnyder, 2009; Robin et al., 2010), (3) the high rates of shoot tissue (mainly leaves) losses that elicit a priority for assimilate (including reserve) allocation to shoot regeneration at the expense of the root system (e.g. Bazot et al., 2005), and (4) predominant placement of the root system near the soil surface dictated by the high need for nutrient interception and uptake (e.g. from excreta deposits), to compensate the high rates of nutrient losses due to grazing (Lemaire et al., 2000). Importantly, (5) in*

*a relatively high number of cases, the model predicted situations in which rainfall recharged mainly the top soil, while SWC at depths >20 cm remained low (e.g. June-end of year 2006, April-October 2007, or May-end of year 2008; Fig. S12; see also below). Principally, however, factors (2)-(4) alone can explain why shallow rooting depth is a typical feature of intensively grazed grasslands (Troughton, 1957; Klapp, 1971). Also, Prechsl …"*

*Further, we added a supplemental figure (Figure S13), showing $\delta^{18}O_{soil}$ with soil depth as predicted by MuSICA (continuous lines) and mean uptake-weighted depth of root water uptake (dashed horizontal lines) on the different sampling dates. Closed circles: observations of $\delta^{18}O_{soil}$ at 7 and 20 cm depth. Sampling date is given by DOY and year, in the lower right corner of each panel:*

[Figure]

Legend of Fig. 3 (P29 L5ff):

*"The $R^2$, MBE and MAE for the relationship between $\delta^{18}O_{stem}$ and the $\delta^{18}O_{soil}$ at 7 cm depth were 0.69, 0.2‰ and 0.7‰ for the observed data (a) and 0.65, –0.2‰ and 0.7‰ for the predicted data (b). Conversely, the $R^2$, MBE and MAE values for the relationship between $\delta^{18}O_{stem}$ and the $\delta^{18}O_{soil}$ at 20 cm depth were 0.34, 1.9‰ and 2.1‰ for the observed data (a) and 0.17, 1.8‰ and 1.9‰ for the predicted data (b)."*

**MB 5**

Also, a total water balance is always a good means of validation and would be nice to have.

We agree with the reviewer. Unfortunately, we could not do a total water balance. E.g. we did not measure runoff (which was probably close to nil in this non-sloping pasture) and ground water recharge. The latter would have required installation of lysimeters, which was impractical on this intensively managed pasture. However, we did validate the model with latent heat flux data that were available from an eddy covariance station at the site, and we assessed the model's performance in predicting total plant-available water in the entire top soil by comparison with plant-available soil water modelling and data for the same site presented in Schnyder et al. 2006.

In the revision, we added a paragraph (P5 L23ff) stating: *"The model was validated with latent energy flux (LE) data obtained from an eddy covariance station (EC) at the site. According to that comparison (Fig. S1), MuSICA estimates were unbiased ($LE_{MuSICA} = 0.997\ LE_{EC}$; $R^2 = 0.59$). Further, we compared MuSICA predictions of total plant-available soil water (PAW, mm) in the entire top soil with PAW modelling and data for the same site presented in Schnyder et al. (2006). For the 2007-2012 data, this yielded the relationship $PAW_{MuSICA} = 0.99\ PAW_{Schnyder\ et\ al.\ 2006} + 7.8$ ($R^2$ 0.83)."*

**MB 6**

The results section contains a lot of discussion (see detailed comments)

We eliminated discussion from the Results section following closely the reviewer's suggestions (see our answers to the specific comments below).

**MB 7**

Why was model not calibrated?

(This question is connected with point MB 5; see response above) We agree that we did not perform a classical calibration in the sense that the different model parameter values were statistically optimised. To do that we would have needed a greater number of hydrological measurements that we did not have (e.g. the dynamics of ground water recharge and soil water contents). The only instance where we did use parameter optimization (fine tuning) was in the case of the factors controlling $^{18}O$ enrichment of leaf water: mesophyll water content and night-time and minimal stomatal conductance (P9 L7-9), as well as the fraction of unenriched water in bulk leaf water. All other parameter values were based on measurements at the site, or – if such measurements were unavailable – on data from literature (as we explain). In that way we did ascertain realistic parameter values in this (otherwise) purely physically-based model. The fact that the model predicted well the $\delta^{18}O_{soil}$ at two different depths (that is a depth within the zone of most active root water uptake, 7 cm, and a depth just below that zone, 20 cm) did indicate strongly that the ensemble of parameters dictating soil water dynamics (including the spatial distribution of soil water uptake) in the zone of water uptake was described well by the model. This conclusion is further substantiated by the sensitivity analysis.

In the revision, we added the following short paragraph (see also response to MB 5) in P14 L15ff: *"The ability of the model to generate realistic predictions of the $\delta^{18}O$ dynamics at different depths in the soil (within the zone of most active root water uptake and just below that zone) suggests strongly that the ensemble of parameters dictating the spatio-temporal dynamics of soil water contents (including emptying and refilling dynamics) was described well in the model. That interpretation was also supported by the sensitivity analysis."*

**MB 8**

Why was 2H not used? How was fractionation evaluated without 2H - did the authors simply use the offset of 18O from the LMWL? Is the model capable of modeling 2H as well? The dual-isotope space enables a more comprehensive understanding of processes. Also, it is more sensitive compared to 18O and since the authors did a sensitivity study, perhaps very useful. I don't say I expect that in a revised version, but I am interested on the authors opinion on that.

Yes, the MuSICA model is capable of simulating the $\delta^2H$ of soil water, xylem and leaf water. However, we elected to not include those data in the manuscript, as (1) we are primarily interested in the processes leading up to the $\delta^{18}O$ of cellulose, (2) we had noticed discrepancies in the model-data agreement for D/H that indicated fractionation (including a surface effect on D/H of soil water at the experimental site; Chen et al., 2016) that are currently not accounted for in the model. Hence, reporting both $\delta^{18}O$ and $\delta^2H$ would have changed the focus of the paper and would have brought up additional questions (that we wish to investigate in a separate paper). Also (3), we did not want to overload the paper with extra figures and discussion.

In the revisions we added the following sentence (P5 L27ff): *Although the MuSICA model is capable of simulating $\delta^2H$ of water pools in the soil-plant system, we excluded those data in the manuscript, as (1) we are primarily interested in the processes leading up to the $\delta^{18}O$ of cellulose, (2) we had noticed discrepancies in the model-data agreement for D/H indicating fractionation (including a surface effect on D/H of soil water at the experimental site; Chen et al., 2016) that are currently not accounted for in the model, and (3) we did not want to overload the paper with extra figures and discussion. Issues of D/H fractionation of water including data from this experimental site will be addressed in a separate paper.*

**MB 9**
Having that said, I suggest minor revision. I am looking forward to see the manuscript published in HESS.

Detailed comments:
Abstract l.20: grazing pressure, but how about rooting depth? Grasses are shallow-rooted so any other uptake is not expected?!

As we mention above, the potential range of rooting depths of perennial grasses (and forbs) is very large and dependent on a wide range of factors including site conditions, species and management conditions (particularly grazing pressure or defoliation frequency). So, the predominance of water uptake from shallow depths is not necessarily a universal feature of grassland.

In the revision we added a phrase in the Abstract, P1 L20ff
"The model accurately predicted the $\delta^{18}O$ dynamics of the different ecosystem water pools, *suggesting that the model generated realistic predictions of the vertical distribution of soil water and root water uptake dynamics. Observations and model predictions indicated that water uptake occurred predominantly from shallow (<20 cm) soil depths ...*"

See also the detailed response to MB 4, above)

**MB 10**
l.20: respond to atmospheric moisture….does that mean leaves take up moisture from the atmosphere? (foliar uptake???)

Yes. Leaves exhibit bidirectional exchange of water vapour with the atmosphere, with a relative magnitude of the inward flux proportional to the relative humidity of the air, as we describe in the manuscript.

In the revision we changed the respective sentence to clarify the fact that it is actually the relative moisture 'content' of the atmosphere that drives the observed relationship. The sentence now reads (P1 L20): "$\Delta^{18}O_{leaf}$ responded to both soil and atmospheric moisture *contents...*"

**MB 11**
l.21: two non-mixing pools: is that realistic or justified?

We see the point. Yes, the idea of two 'non-mixing' pools is a simplification, and unrealistic in the strict sense. The idea of having two discrete water pools in a leaf is the simplest conceptual model for explaining the observation that leaf water is usually less enriched than predicted by the Craig-Gordon model. The two-pool model is based on the notion that xylem and ground tissue are composed of unenriched water, whereas mesophyll cells are filled with evaporatively enriched water, implying constant fractions of unenriched and enriched leaf water (given full hydration of the leaves).
However, the reviewer is correct in questioning the realism of the 'non-mixing pools' idea, particularly in grasses that exhibit a continuous $^{18}O$-enrichment towards the tip.

So, in the revisions we replaced the term 'two non-mixing water pools' by 'two pool' model characterized by constant proportions of unenriched and evaporatively enriched water. In the Abstract, this sentence now reads (P1 L20ff): *"$\Delta^{18}O_{leaf}$ responded to both soil and atmospheric moisture contents and was best described in terms of constant proportions of unenriched and evaporatively enriched water (two-pool model)."*

**MB 12**
l.26: the second sentence is not well written/unconcise

The revised sentence now reads:
*"Meteoric waters impart their isotopic signal ($\delta^{18}O_{rain}$) to that of soil water ($\delta^{18}O_{soil}$), changing it as a function of refilling, exchange and percolation processes throughout the soil profile."*

**MB 13**
l.29: explain better or provide citation – explain why do leaves fractionate

The revised sentence now reads:
*"The oxygen isotope composition of leaf water ($\delta^{18}O_{leaf}$) differs from that of the water taken up from the soil, as leaf water becomes $^{18}O$-enriched due to evaporative effects and morpho-physiological controls (Barbour 2007)."*

**MB 14**
p. 2 l.14: 'source water' for plants would be soil or groundwater, but not xylem water as it is plant water already

We revised the sentence accordingly:
*"The isotopic composition of the water taken up by plants (henceforth termed $\delta^{18}O_{stem}$) can vary over time through changes in the depth of soil water uptake by roots or direct changes in soil water isotopic composition."*

**MB 15**
p. 2 l.15/16: 'summer' and 'winter' should be related to the particular study area, these statements are not true for the whole earth….

We modified the sentence accordingly: *"For example, summer rains in continental Europe are usually isotopically distinct ($^{18}O$-enriched) relative to winter precipitation, generating intra-annual variations of $\delta^{18}O_{soil}$ ($\delta^{18}O$ of soil water) with soil depth."*

**MB 16**

p. 2 l.29: 'enrichment above....' I know what you mean but this is written ambiguous – stem water can also be subject to fractionation under certain conditions. It should be more clearly expressed what is meant with this sentence.

We see the point.
Here we use the term $\delta^{18}O_{stem}$ to denote the $\delta^{18}O$ of the water taken up from the soil, and we define that term on first use. In what follows, we assume that there is no (relevant) further fractionation against $^{18}O$, so that the water entering the leaf has the same $\delta^{18}O$ as that taken up by the root system as a whole.

We revised the annotated sentence, specifying that point: *"The mechanisms driving the isotopic enrichment of leaf water can be studied separately from those driving changes in $\delta^{18}O_{stem}$ by expressing the isotopic composition of leaf water as enrichment above $\delta^{18}O_{stem}$, i.e., $\Delta^{18}O_{leaf} = \delta^{18}O_{leaf} - \delta^{18}O_{stem}$, if the $\delta^{18}O$ of water entering the leaf is the same as that taken up by the root system as a whole.*

**MB 17**
p. 2 l.31: 'many authors' – could you provide some citations, please?

We added a citation to a pertinent review: Cernusak et al. 2016.

**MB 18**
p. 3 ll.2-14: this is well written!

Thank you!

**MB 19**
p. 3 l.15: is this relevant for grasslands only?

Actually, there is no reason to believe that this is only relevant for grassland.

So, we deleted 'grassland'.

**MB 20**
p.4.l.5: please review this sentence and provide more information…which species, which soil depths, what exactly is meant with 'growing season'

We added the requested info.

The paragraph now reads: *"To explore these questions we compared predictions from the $^{18}O$ -enabled soil-plant-atmosphere model MuSICA (Ogée et al., 2003; Wingate et al., 2010; Gangi et al., 2015) with those observed in a unique, multi-annual data set (7 years) of growing season (April to November), biweekly samplings and $\delta^{18}O$ analysis of soil water (at 7 and 20 cm depth), stem and midday leaf water, atmospheric water vapour, along with rainfall amount and $\delta^{18}O_{rain}$ data. The experimental site (Schnyder et al., 2006) was an intensively grazed Lolio-Cynosuretum (Williams and Varley, 1967; Klapp, 1965) community with Lolium perenne, Poa pratensis, Dactylis glomerata, Phleum pratense, Taraxacum officinale, and Trifolium repens as the main species. Vegetation samples were taken as mixed-species samples, as described below.*

**MB 21**
p.5.l8: though you cite a paper on the cryogenic system you use, it would be nice to specify temperature and extraction time here

We revised the sentence as follows:

*"All samples were stored in a freezer at approx. -18°C until water extraction. Water was extracted for two hours using a cryogenic vacuum distillation apparatus with sample vials placed in a water bath with a temperature set to 80°C (Liu et al., 2016)."*

**MB 22**
p.6.ll. 1 & 2-7: These information belong together, I'd suggest to either put the first part down or the second up

We followed the recommendation and revised the paragraph as follows:
*"MuSICA was forced by half-hourly values of meteorological data and $\delta^{18}O$ of water vapour ($\delta^{18}O_{vapour}$) and rainwater ($\delta^{18}O_{rain}$). Wind speed, precipitation, air temperature, relative humidity and air pressure data were obtained from the Munich airport meteorological station, located at about 3 km south of the experimental site. Radiation was calculated as the mean of two weather stations located 10 km west and 12 km east of the experimental site. $CO_2$ concentration was measured at the site by an open-path infrared $CO_2/H_2O$ gas analyser (LI-7500, LI-Cor, Lincoln, USA). For $\delta^{18}O_{vapour}$ and $\delta^{18}O_{rain}$, observations at the experimental site were used whenever available. Otherwise $\delta^{18}O_{vapour}$ and $\delta^{18}O_{rain}$ estimates were obtained from globally-gridded reconstructions derived from the isotope-enabled, nudged atmospheric general circulation model IsoGSM (Yoshimura et al., 2011). The IsoGSM-predicted $\delta^{18}O_{vapour}$ and $\delta^{18}O_{rain}$ at the grid point relevant to our site were first corrected for their offset with observed data, as predictions were found to be more enriched by 2‰ and 1.3‰ on average compared to the $\delta^{18}O_{vapour}$ and $\delta^{18}O_{rain}$ measured at the site (Figs. S2–S4)."*

**MB 23**
p.7 l. 33: based on what was the beta distribution assumed (based on previous research or citation)

The beta distribution was shown to provide a good description of the vertical distribution of root-length-densities (e.g. Sadri et al., 2018).

We added a reference to Sadri et al. (2018).

**MB 24**
p.10.l 2: Why does the ratio need to remain 1.6?

In their review, Medlyn *et al.* (2002) found a close relationship between the potential rate of electron transport ($J_{max}$) and the maximum rate of carboxylation ($V_{cmax}$) for a broad range of crop, broadleaf and coniferous species. The slope of that regression was 1.6. Based on that study, we assumed a constant $J_{max}/V_{cmax} = 1.6$ also in our work (see Supplement, Table S1).

In the revision, we added the citation to Medlyn et al. (2002) in the main text. The sentence now reads:
*"$V_{cmax}$ and $J_{max}$ were altered in tandem to keep the ratio $J_{max}/V_{cmax}$ at 1.6 (Medlyn et al., 2002), the same as in the standard simulation (Table S1)."*

**MB 25**
p.10.ll. 4-6: Perhaps that fits better to 2.4.1 isoforcing

We revised the text in section 2.5 that was misleading, to clarify that the sentence relates to the sensitivity analysis and not to the isoforcing for the standard simulation.

That sentence now reads *"In addition, we investigated the effect of using uncorrected IsoGSM-predicted $\delta^{18}O_{rain}$ and $\delta^{18}O_{vapour}$ data instead of local isotopic data (gap-filled with offset-corrected IsoGSM data; see 2.4.1) for the isoforcing of MuSICA. This served to illustrate the usefulness of having local rainwater $\delta^{18}O$ data."*

**MB 26**
p.10.l21: Was predicted soil water content validated somehow?

Yes, we obtained a good agreement between predictions of soil water content with MuSICA with predictions obtained using the approach described by Schnyder et al. (2006) for the same site.

See response to MB 5, above

**MB 27**
p.11. l 29: in the way that (word missing)

We inserted '*in the way that*'.

**MB 28**
p.11: paragraph 3.4 contains a lot of discussion, I suggest reviewing and removing some of the 'judging' (e.g. last sentence or l.29/30)

We revised the paragraph, accordingly.

**MB 29**
p.12.l.21: MLR does not appear in the methods/statistics

We added in the Statistics section: *"Simple and multiple linear regression analyses and student's t tests were performed in R, version 3.4.2 (R Core Team, 2017) and RStudio, version 1.1.383 (RStudio Team, 2016)."*

**MB 30**
p.12.l.23: weakly significant? I think this should be rephrased ! significant or not

The P values for the predicted and observed regressions lay between 0.05 and 0.1, i.e. close to significant. Thus, the sentence was rephrased as follows: *"The interaction effect of air relative humidity and SWC was close to significant for both observed (P = 0.080) and predicted (P = 0.073) $\Delta^{18}O_{leaf}$ (Table 4)."*

**MB 31**
p.12. paragraph 3.5.: the authors mix VPD and relative humidity quite a lot here, which makes this chapter hard to read. I suggest restructuring and rephrasing of this chapter (though the results completely make sense)

We agree and restructured the paragraph.

The new text now reads: *"Multiple regression analysis demonstrated significant effects of air relative humidity (P < 0.01) and SWC (P < 0.05) on both observed and predicted $\Delta^{18}O_{leaf}$ (Table 4). $\Delta^{18}O_{leaf}$ increased with decreasing air relative humidity and SWC (Figs. 4a, b and 5a, b). The interaction effect of air relative humidity and SWC was close to significant for both observed (P = 0.080) and predicted (P = 0.073) $\Delta^{18}O_{leaf}$ (Table 4). The effect of dry soil conditions on $\Delta^{18}O_{leaf}$ was most evident at low air humidity (Figs. 4a, b and 5a, b) and was connected with a decrease of canopy conductance ($g_{canopy}$) (Fig. 5c).*
*The modelled dependence of transpiration on air VPD (the climatic driver of transpiration) was strongly modified by SWC (Fig. 4c). High air VPD drove high transpiration rates only under wet soil conditions (SWC ≥ 0.25)."*

**MB 32**
p.13l 4-10: Discussion
p.13. l.26-32: This sounds more like a conclusion

This paragraph is summarizing the main observations on model-data agreement. We would like to retain it, as it is.

**MB 33**

p.14. l.5: quite

We removed 'quite'

**MB 34**

p.14. l.6-7: suggest rephrasing: 'likely result from sampling effects and analytical error'

We agree and rephrased the sentence as follows: "*The greater scatter in the observed relationship between $\Delta^{18}O_{leaf}$ and relative humidity compared to predictions (Fig. 4) likely resulted partly from sampling effects and error.*"

**MB 35**

p.14. l.12-23: I agree, but also it should be clear that grass with a fairly uniform uptake depth right below surface is probably the easiest of plants to model. This is not a criticism but would be interesting how the model performs for different plant types.

We agree, in principle. Yes, it would be extremely interesting to also test the model for its performance with different biomes in different site conditions, exploring also especially systems that include deep-rooted species.

**MB 36**

4.2: I am not sure if this deserves an own chapter. I believe that it is true that the grass takes the water mainly from the upper depths but considering the characteristic shape of soil water isotope profiles at the surface (enrichment and subsequent decrease of isotope values towards a constant value), the used resolution of only 2 depths might not reveal true uptake patterns. Also see Rothfuss and Javaux, 2016.

We see the point, and the caveat. We are aware of the fact that the soil water $\delta^{18}O$ values from only two depth positions do not necessarily reflect the total range of $\delta^{18}O$ expected for the entire soil profile. Nevertheless, the model simulations generated a detailed prediction of how $\delta^{18}O$ varied along the profile. For the sampled depth, the predictions matched the observations generally well. We added a supplemental figure (Figure S13) showing the predicted soil water $\delta^{18}O$ profiles (see response to MB 4, above). The most extreme (positive) values were predicted for the uppermost 1-2 cm of the soil (Fig. S13), as a consequence of evaporative $^{18}O$ enrichment at the soil surface. The model predicted very little root water uptake in that zone (Fig. S12).

The $\delta^{18}O$ of soil water at 7 cm was greater (i.e. more enriched) than the $\delta^{18}O$ at 20 cm for 79 out of 86 cases, i.e. for more than 90% of the dataset. In line with that, the model mostly predicted a decrease of $\delta^{18}O$ between 7 and 20 cm, which was monotonous for a large part of the dataset (new Figure S13). Even if the decrease was not monotonous (e.g. in late summer/autumn of 2006), the highest and lowest $\delta^{18}O$ values were still found in the upper and lower profile, respectively. Hence, at least the qualitative assessment that the roots take up the water from the shallow horizon was still valid in those cases.

On 12 days, $\delta^{18}O_{soil}$ was predicted to be quite constant from approx. 5 cm to the bottom of the profile. In those specific cases, additional soil samples between 5 and 37 cm would not have had additional value with regard to inferring the depth of water uptake by comparing $\delta^{18}O_{stem}$ and $\delta^{18}O_{soil}$. On another 6 days in 2008 and 2010 (e.g. DOY 209 and 285 in 2010), the uptake depth could not be unambiguously inferred by comparing $\delta^{18}O_{stem}$ and $\delta^{18}O_{soil}$. Considerable rainfall had occurred in the two weeks preceding those sampling days (e.g. 61 litres of rain during DOY 203 to 208 of 2010), creating non-monotonous isotopic profiles (e.g. an S-shaped profile on DOY 209 of 2010). In those cases, the model predictions, which were solely based on hydraulic properties of the soil, root architecture and evaporative demand, and not on observed $\delta^{18}O_{soil}$ data, can help to deduce the root water uptake depth. For day 209 in 2010 for example, the model predictions indicated that the average mass-weighted root water uptake depth was located at 10.5 cm (dashed horizontal line in Fig. S13 for that DOY).

We revised this chapter thoroughly, paying close attention to the reviewers' concerns. See response to MB 4, above.

**MB 37**
p.15. l.26-27: 'online transpiration isotope method' this appears here for the first time?

Yes. These data help us in the discussion, in that they provide supporting evidence for the two-pool model also for individual grass species (that were part of the codominant species in our grassland community).

The methods and results of these supplementary experiments with *Lolium perenne* and *Dactylis glomerata* are described in the Supplement. The citation to that description (Notes S2) was missing and is now added to the revised manuscript:
*"We did not know if putative between-species differences in leaf water dynamics and associated $^{18}$O-enrichment, or any other morpho-physiological effects e.g. associated with leaf aging, could have led to a missing correlation between the proportional difference between measured leaf water $^{18}$O-enrichment and that predicted by the Craig-Gordon model (1 - $\Delta^{18}O_{leaf}/ \Delta^{18}O_e$) and transpiration rate. For these reasons, we explored this question with separate studies of L. perenne and D. glomerata, two species that also formed part of the present grazed grassland ecosystem. Again, these studies found no evidence for a Péclet effect, and supported the two-pool model, as there was* no relationship between the proportional difference between measured leaf water enrichment and that predicted by the Craig-Gordon model (1 - $\Delta^{18}O_{leaf}/ \Delta^{18}O_{e,ss}$) and transpiration rate in either *L. perenne* plants grown in a controlled environment at different relative humidities and water availabilities, or *D. glomerata* leaves measured using an online transpiration isotope method *(Notes S2 and Figs. S14-15)."*

**MB 38**
p.16 l.9-11: I like this chapter, but the last sentence does not make sense – why compare and justify grass species with a study on non-grass-species?

We do not wish to justify our data by comparison with non-grass species. However, it is interesting and important to note that the range of proportional differences between measured leaf water $^{18}$O enrichment and that predicted by the Craig-Gordon model ($\varphi$) is very similar in grasses and dicots.
We revised the faulted sentence, which now reads: *"Considering a similar effect of vein removal would move our observed $\varphi$ to about 0.2. Such a value of $\varphi$ for grasses is very similar to the mean $\varphi$ reported for a wide range of non-grass species by Cernusak et al. (2016)."*

**MB 39**
Conclusions: An experienced and known Professor once gave me the advice 'A good paper doesn't need a conclusion – the reader draws it him/herself.' The authors should decide themselves, but I feel emphasizing some key points in the manuscript/abstract a bit more would be sufficient without conclusion.

We deleted the Conclusions, and emphasized key points, as documented above.

**MB 40**
Fig. 3: As stated above, the model does not work that well for 18O. I think this needs to be discussed thoroughly

See our response to MB 4 (above).

**References not included in the Discussion paper**

Auerswald, K. and Schnyder, H.: Böden als Grünlandstandorte, in: Handbuch der Bodenkunde, edited by: Blume, H.-P., Felix-Henningsen, P., Frede, H.-G., Guggenberger, G., Horn, R., and Stahr, K., Wiley-VCH, 31, Erg.Lfg., 1-15, https://doi:10.1002/9783527678495.hbbk2009003, 2009.

Bazot, S., Mikola, J., Nguyen, C., and Robin, C.: Defoliation-induced changes in carbon allocation and root soluble carbon concentration in field-grown *Lolium perenne* plants: do they affect carbon availability, microbes and animal trophic groups in soil?, Funct. Ecol., 19, 886-896, https://doi.org/10.1111/j.1365-2435.2005.01037.x, 2005.

Brinkmann, N., Seeger, S., Weiler, M., Buchmann, N., Eugster, W., and Kahmen, A.: Employing stable isotopes to determine the residence times of soil water and the temporal origin of water taken up by *Fagus sylvatica* and *Pices abies* in a temperate forest, New Phytol., 219, 1300-1313, https://doi.10.1111/nph.15255, 2018.

Chen, G., Auerswald, K., and Schnyder, H.: $^2$H and $^{18}$O depletion of water close to organic surfaces, Biogeosciences, 13, 3175-31186, https://doi:10.5194/bg-13-3175-2016, 2016.

Klapp, E.: Grünlandvegetation und Standort, Parey, Berlin, 1965.

Lemaire, G., Hodgson, J., de Moraes, A., and Nabinger, C.: Grassland Ecophysiology and Grazing Ecology, CABI Publishing, Wallingford, U.K., 2000.

Medlyn, B. E., Dreyer, E., Ellsworth, D., Forstreuter, M., Harley, P. C., Kirschbaum, M. U. F., Le Roux, X., Montpied, P., Strassemeyer, J., Walcroft, A., Wang, K., and Loustau, D.: Temperature response of parameters of a biochemically based model of photosynthesis. II. A review of experimental data, Plant Cell Environ., 25, 1167–1179, https://doi.org/10.1046/j.1365-3040.2002.00891.x, 2002.

Robin, A. H. K., Matthew, C., and Crush, J. R.: Time course of root initiation and development in perennial ryegrass – a new perspective, Pr. N. Z. Grassl. Assoc., 72, 233-240, 2010.

Rothfuss, Y. and Javaux, M.: Review and syntheses: Isotopic approaches to quantify root water uptake: a review and comparison of methods, Biogeosciences, 14, 2199-2224, https://doi:10.5194/bg-14-2199-2017, 2017.

Sadri, S., Wood, E. F., and Pan, M.: Developing a drought-monitoring index for the contiguous US using SMAP, Hydrol. Earth Syst. Sc., 22, 6611-6626, https://doi.org./10.5194/hess-22-6611-2018, 2018.

Schenk, H. J. and Jackson, R.B.: Rooting depths, lateral root spreads and below-ground/above-ground allometries of plants in water-limited ecosystems, J. Ecol., 90, 480-494, https://doi.org/10.1046/j.1365-2745.2002.00682.x, 2002.

Williams, J. T. and Varley, Y. W.: Phytosociological studies of some British grasslands. I. Upland pastures in Northern England, Vegetatio 15, 169-189, https://doi.org./10.1007/BF01963747, 1967.

Yang, J. Z., Matthew, C., and Rowland, R. E.: Tiller axis observations for perennial ryegrass (*Lolium perenne*) and tall fescue (*Festuca arundinacea*): number of active phytomers, probability of tiller appearance, and frequency of root appearance per phytomere for three cutting heights, New Zeal. J. Agr. Res., 41, 11-17, https://doi:10.1080/00288233.1998.9513283, 1998.